# LEARNING RIGID DYNAMICS WITH FACE INTERACTION GRAPH NETWORKS

**Kelsey R. Allen**\*, **Yulia Rubanova**\*, **Tatiana Lopez-Guevara**, **William Whitney**,
**Alvaro Sanchez-Gonzalez**, **Peter Battaglia**, **Tobias Pfaff**
DeepMind, London, UK

## ABSTRACT

Simulating rigid collisions among arbitrary shapes is notoriously difficult due to complex geometry and the strong non-linearity of the interactions. While graph neural network (GNN)-based models are effective at learning to simulate complex physical dynamics, such as fluids, cloth and articulated bodies, they have been less effective and efficient on rigid-body physics, except with very simple shapes. Existing methods that model collisions through the meshes' nodes are often inaccurate because they struggle when collisions occur on faces far from nodes. Alternative approaches that represent the geometry densely with many particles are prohibitively expensive for complex shapes. Here we introduce the "Face Interaction Graph Network" (FIGNet) which extends beyond GNN-based methods, and computes interactions between mesh *faces*, rather than nodes. Compared to learned node- and particle-based methods, FIGNet is around 4x more accurate in simulating complex shape interactions, while also 8x more computationally efficient on sparse, rigid meshes. Moreover, FIGNet can learn frictional dynamics directly from real-world data, and can be more accurate than analytical solvers given modest amounts of training data. FIGNet represents a key step forward in one of the few remaining physical domains which have seen little competition from learned simulators, and offers allied fields such as robotics, graphics and mechanical design a new tool for simulation and model-based planning.

## 1 INTRODUCTION

Simulating rigid bodies accurately is vital in a wide variety of disciplines from robotics to graphics to mechanical design. While popular general-purpose tools like Bullet (Coumans, 2015), MuJoCo (Todorov et al., 2012) and Drake (Tedrake, 2019) can generate plausible predictions, predictions that match real-world observations accurately are notoriously difficult (Wieber et al., 2016; Anitescu & Potra, 1997; Stewart & Trinkle, 1996; Fazeli et al., 2017; Lan et al., 2022). Numerical approximations necessary for efficiency are often inaccurate and unstable. Collision, contact and friction are challenging to model accurately, and hard to estimate parameters for. The dynamics are non-smooth and nearly discontinuous (Pfrommer et al., 2020; Parmar et al., 2021), and influenced heavily by the fine-grained structure of colliding objects' surfaces (Bauza & Rodriguez, 2017). Slight errors in the physical model or state estimates can thus lead to large errors in objects' predicted trajectories. This underpins the well-known sim-to-real gap between results from analytical solvers and real-world experiments.

Learned simulators can potentially fill the sim-to-real gap. They can be trained to correct imperfect state estimation, and can learn physical dynamics directly from observations, potentially producing more accurate predictions than analytical solvers (Allen et al., 2022; Kloss et al., 2022). Graph neural network (GNN)-based models, in particular, are effective at simulating liquids, sand, soft materials and simple rigids (Sanchez-Gonzalez et al., 2020; Mrowca et al., 2018; Li et al., 2019b; Pfaff et al., 2021; Li et al., 2019a). Many GNN-based models are node-based: they detect and resolve potential collisions based on whether two mesh nodes or particles are within a local neighborhood. However, collisions between objects do not only happen at nodes. For example, two cubes may

---

\*These authors contributed equally. Correspondence to {`krallen`,`rubanova`}@deepmind.com

collide by one corner hitting the other's face, or one edge hitting the other's edge (in fact, corner-to-corner collisions are vanishingly rare). For larger meshes, fewer collisions occur within the local neighborhood around nodes, and thus collisions may be missed. Some prior work thus restricts collisions to simple scenarios (e.g. a single object colliding with a floor (Allen et al., 2022; Pfrommer et al., 2020)). Alternative approaches represent the object densely with particle nodes (Li et al., 2019a; Sanchez-Gonzalez et al., 2020), but that leads to a quadratic increase in node-node collision tests, which is computationally prohibitive for nontrivial scenes.

Here we introduce a novel mesh-based approach to collision handling—Face Interaction Graph Networks (FIGNet)—which extends message passing from graphs with directed edges between nodes, to graphs with directed hyper-edges between faces. This allows FIGNet to compute interactions between mesh faces (whose representations are informed by their associated nodes) instead of nodes directly, allowing accurate and efficient collision handling on sparse meshes without missing collisions. Relative to prior node-based models, we show that for simulated multi-rigid interaction datasets like Kubric (Greff et al., 2022), FIGNet is 8x more efficient at modeling interactions between sparse, simple rigid meshes, and around 4x more accurate in translation and rotation error for predicting the dynamics of more complex shapes with hundreds or thousands of nodes. We additionally show that FIGNet even outperforms analytical solvers for challenging real-world robotic pushing experiments (Yu et al., 2016). To our knowledge, FIGNet is the first fully learned simulator that can accurately model collision interactions of multiple rigid objects with complex shapes.

## 2 RELATED WORK

To address the weaknesses of traditional analytic simulation methods, a variety of hybrid approaches that combine machine learning with analytic physics simulators have been proposed. Learned simulation models can be used to correct analytic models for a variety of real-world domains (Fazeli et al., 2017; Ajay et al., 2018; Kloss et al., 2017; Golemo et al., 2018; Zeng et al., 2020; Hwangbo et al., 2019; Heiden et al., 2021). Analytic equations can also be embedded into neural solvers that exactly preserve physical laws (Pfrommer et al., 2020; Jiang et al., 2022). While hybrid approaches can improve over analytic models in matching simulated and real object trajectories, they still struggle in cases where the analytic simulator is not a good model for the dynamics, and cannot be trivially extended to non-rigid systems.

More pure learning-centric approaches to simulation have been proposed in recent years to support general physical dynamics, and GNN-based methods are among the most effective. GNNs represent entities and their relations with graphs, and compute their interactions using flexible neural network function approximators. Such methods can capture the dynamics of fluids and deformable meshes (Li et al., 2019a; Sanchez-Gonzalez et al., 2020; Pfaff et al., 2021), robotic systems (Pathak et al., 2019; Sanchez-Gonzalez et al., 2018; Wang et al., 2018), and simple rigids (Rubanova et al., 2021; Mrowca et al., 2018; Li et al., 2019b; Battaglia et al., 2016; Allen et al., 2022; Bear et al., 2021).

Despite these models' successes in many areas, the rigid dynamics settings they have been applied to generally use very simple shapes like spheres and cubes. Furthermore, most models are tested only in simulation, which may be a poor proxy for real world rigid dynamics (Bauza & Rodriguez, 2017; Fazeli et al., 2017; Acosta et al., 2022). It is therefore unclear whether end-to-end deep learning models, including GNNs, are generally a good fit for learning rigid-body interactions, particularly if the rigid bodies are complex, volumetric objects like those we experience in the real world.

In this work, we demonstrate how using insights from graphics for representing and resolving collisions as face-to-face interactions improves rigid body dynamics learning enough to not only very accurately capture simulated dynamics, but also outperform analytic simulators (Todorov et al., 2012; Coumans, 2015; Lynch, 1992) for real-world rigid-body dynamics data. We conduct a series of ablations and experiments which showcase how face-to-face collision representations dramatically improve rigid body dynamics prediction.

## 3 METHOD

For the purposes of rigid-body simulation, objects are represented as meshes $M$, consisting of the set of node positions $\{\mathbf{x}_i\}_{i=1..N}$ and a set of faces $\{\mathcal{F}\}$ that describe how nodes are connected to

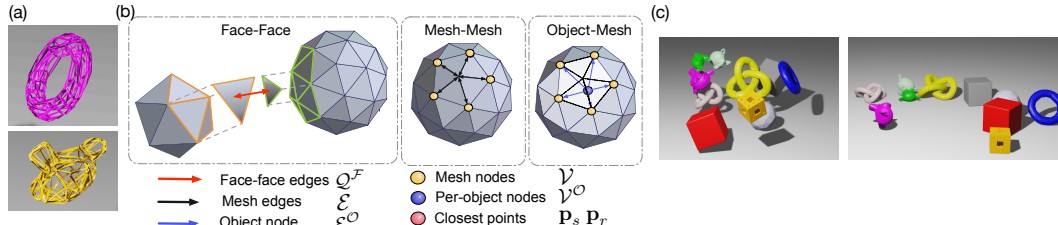

Figure 1: FIGNet model with face-to-face edges. (a) Example meshes from the Kubric MOVi-B dataset (Greff et al., 2022). (b) FIGNet exploits three types of interactions: face-face collisions between objects, mesh-mesh interactions within an object, and interactions with the virtual per-object node. (c) Rollout from FIGNet trained on Kubric MOVi-B (Greff et al., 2022).

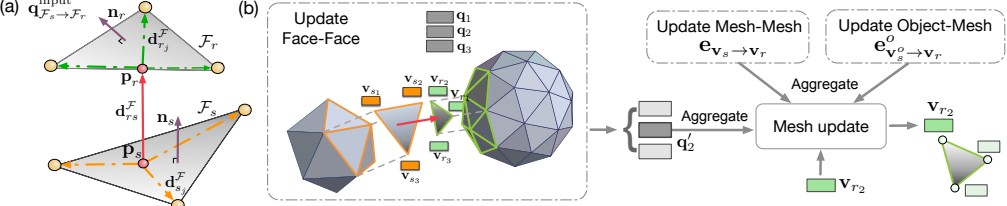

Figure 2: (a) Calculating face-face collision edge features (b) Message-passing with face-face edges. Face-face edges produce the updates for each node of the face (grey rectangles). After aggregation, the face-face update for the corresponding face node is combined with Mesh-Mesh and Object-Mesh edge representations, similarly to standard GNN message-passing. Notation is defined in text.

form objects in a mesh. Each face consists of three nodes $\mathcal{F} = \{s_j\}_{j=1,2,3}$, where $s_j$ are the indices of the nodes in the face. We consider triangular faces but all aspects of the method generalize to faces with other numbers of nodes (quads, tetrahedra, points, segments, etc.) and their interactions.

The simulation trajectory is represented as a sequence of meshes $\mathcal{M} = (M^{t_0}, M^{t_1}, M^{t_2}, \dots)$, where each mesh represents one of more objects. It is computed by applying a sequence of object rotations and translations to a reference static mesh $M^U$ of each object. The goal of the simulator $s$ is to predict the next state of the system $\tilde{M}^{t+1}$ from a history of previous states $\{M^t, ..., M^{t-h+1}\}$. We use a history length of $h = 2$, following (Allen et al., 2022). The rollout trajectory can be obtained by iteratively applying the simulator to the previous prediction; $(M^{t_0}, \tilde{M}^{t_1}, \dots, \tilde{M}^{t_k})$.

## 3.1 FACE INTERACTION GRAPH NETWORK

We introduce the Face Interaction Graph Network (FIGNet) (Figs. 1 and 2) as the simulator $s$, which is designed to extend existing general purpose simulators (Sanchez-Gonzalez et al., 2020; Pfaff et al., 2021) to model interactions between multiple rigid objects more efficiently. FIGNet is an evolution of the Encode-Process-Decode architecture which performs message passing on the nodes and edges of the object meshes, adding two key extensions over MeshGraphNets(Pfaff et al., 2021). First, to model collisions between two objects, FIGNets generalizes message passing over directed edges to message passing over directed hyper-edges. This enables message passing on face-face edges, from a face with three sender nodes to a face with three receiver nodes. FIGNet replaces MeshGraphNets'(Pfaff et al., 2021) node-based "world edges" with face-face edges when faces from distinct objects are within a certain distance $d_c$. Second, FIGNet supports multiple node and edge types. We use this to add "object nodes" that are connected to all of the mesh nodes of a given object. These also participate in message passing (Fig. 2b) to enable long-range communication[1]

### 3.1.1 ENCODER

The encoder constructs the input graph $\mathcal{G} = (\mathcal{V}, \mathcal{E}, \mathcal{Q}^{\mathcal{F}})$ from the mesh $M^t$. The vertices $\mathcal{V} = \{v_i\}$ represent the mesh nodes, $\mathcal{E} = \{e_{v_s \rightarrow v_r}\}$ are bidirectional directed edges added for all pairs of nodes that are connected in the mesh, and $\mathcal{Q}^F = \{q_{\mathcal{F}_s \rightarrow \mathcal{F}_r}\}$ are edges between the *faces* of different

---

[1]For simplicity on the notation we will omit "object nodes" in the explanation of the model in the main text.

objects that are close spatially. Each edge in $\mathcal{Q}^{\mathcal{F}}$ is defined between sets of nodes of the sender face $\mathcal{F}_s = (v_{s1}, v_{s2}, v_{s3})$ and the receiver face $\mathcal{F}_r = (v_{r1}, v_{r2}, v_{r3})$.

**Node features and mesh edges**  We follow (Pfaff et al., 2021) for constructing the features for each mesh node $v_i$ and mesh edge $e_{v_s \to v_r}$. The features of $v_i$ are the finite-difference velocities estimated from the position history $\mathbf{v}_i^{\text{features}} = [\mathbf{x}_i^t - \mathbf{x}_i^{t-1}, \mathbf{x}_i^{t-h+1} - \mathbf{x}_i^{t-h}]$, where [] indicates concatenation. The features of $e_{v_s \to v_r}$ are computed as $\mathbf{e}_{v_s \to v_r}^{\text{features}} = [\mathbf{d}_{rs}, \mathbf{d}_{rs}^U]$, where $\mathbf{d}_{rs} = \mathbf{x}_s - \mathbf{x}_r$ is the relative displacement between the nodes in the current mesh $M^t$ and $\mathbf{d}_{rs}^U$ is the relative displacement in the reference mesh $M^U$. Note that this representation of the graph is translation-invariant, as it does not contain the absolute positions $\mathbf{x}_i$, leading to a translation equivariant model with better generalization properties (Pfaff et al., 2021). Finally, we use separate MLP encoders to embed the features of each mesh node, and mesh edge into latent representations $\mathbf{v}_i$ and $\mathbf{e}_{v_s \to v_r}$.

**Face-face edge features**  To model face-face collisions between two rigid objects, we introduce a new type of face-face edges, $\mathcal{Q}^{\text{F}}$. A face-face edge $q_{\mathcal{F}_s \to \mathcal{F}_r}$ connects a sender face $\mathcal{F}_s = (v_{s1}, v_{s2}, v_{s3})$ to a receiver face $\mathcal{F}_r = (v_{r1}, v_{r2}, v_{r3})$. To determine which faces to connect, we use the Bounding Volume Hierarchy (BVH) algorithm (Clark, 1976) to find all pairs of faces from different objects that are within a certain pre-defined collision radius $d_c$.

To construct face-face edge features (Fig. 2a) we compute the closest points between the two faces, $p_s$ and $p_r$ for faces $\mathcal{F}_s$ and $\mathcal{F}_r$ respectively. Geometrically, the closest point for the sender/receiver face might be either inside of the face, on one of the face edges, or at one of the nodes. Using the closest points as local coordinate systems for each of the faces involved in the collision, we build the following translation equivariant vectors: (1) the relative displacement between the closest points $\mathbf{d}_{rs}^{\mathcal{F}} = \mathbf{p}_r - \mathbf{p}_s$ at the two faces, (2) the spanning vectors of three nodes of the sender face $\mathcal{F}_s$ relative to the closest point at that face $\mathbf{d}_{s_i}^{\mathcal{F}} = \mathbf{x}_{s_i} - \mathbf{p}_s$, (3) the spanning vectors of three nodes of the receiver face $\mathcal{F}_r$ relative to the closest point at that face $\mathbf{d}_{r_i}^{\mathcal{F}} = \mathbf{x}_{r_i} - \mathbf{p}_r$, and (4) the face normal unit-vectors of the sender and receiver faces $\mathbf{n}_s$ and $\mathbf{n}_r$. These vectors are the equivalent to the relative displacement for $\mathbf{d}_{rs} = \mathbf{x}_s - \mathbf{x}_r$ in regular edges, and provide a complete, fully-local coordinate system while keeping the model translation equivariant. In summary:

$$\mathbf{q}_{\mathcal{F}_s \to \mathcal{F}_r}^{\text{features}} = [\mathbf{d}_{rs}^{\mathcal{F}}, [\mathbf{d}_{s_j}^{\mathcal{F}}]_{j=1,2,3}, [\mathbf{d}_{r_j}^{\mathcal{F}}]_{j=1,2,3}, \mathbf{n}_s, \mathbf{n}_r] \tag{1}$$

We use an MLP followed by a reshape operation to encode **each face-face edge as three latent face-face edge vectors**, one for each receiver node:

$$Q_{\mathcal{F}_s \to \mathcal{F}_r} = [\mathbf{q}_{j, \mathcal{F}_s \to \mathcal{F}_r}]_{j=1,2,3} = \text{reshape}(\text{MLP}_{\mathcal{Q}^{\text{F}}}^{\text{encoder}}(\mathbf{q}_{\mathcal{F}_s \to \mathcal{F}_r}^{\text{features}}))) \tag{2}$$

For each face-face edge, and before computing features (2) and (3), we always sort the nodes of the sender face and the receiver face as function of the distance to their respective closest points. This achieves permutation equivariance of the entire model w.r.t. the order in which the sender and receiver nodes of each face are specified.

### 3.1.2  PROCESSOR

The procesor is a GNN generalized to directed hyper-edges that iteratively applies message passing across the mesh edges and face-face edges. We use 10 iterations of message passing with unshared learnable weights. Below we describe a single step of message passing to update the edge latents $\mathbf{e}_{v_s \to v_r}$, the face-face edge latents $Q_{\mathcal{F}_s \to \mathcal{F}_r}$, and the node latents $\mathbf{v}_i$, and produce $\mathbf{e}'_{v_s \to v_r}$, $Q'_{\mathcal{F}_s \to \mathcal{F}_r}$, and $\mathbf{v}'_i$.

The mesh edge latents $\mathbf{e}_{v_s \to v_r}$ are updated following the standard edge update approach from (Battaglia et al., 2018):

$$\mathbf{e}'_{v_s \to v_r} = \text{MLP}_{\mathcal{E}}^{\text{processor}}([\mathbf{e}_{v_s \to v_r}, \mathbf{v}_s, \mathbf{v}_r]) \tag{3}$$

The approach to update each face-face edge is similar, except that each face-face edge has 3 latent vectors, 3 sender mesh nodes and 3 receivers mesh nodes; and produces 3 output vectors:

$$Q'_{\mathcal{F}_s \to \mathcal{F}_r} = [\mathbf{q}'_{j, \mathcal{F}_s \to \mathcal{F}_r}]_{j=1,2,3} = \text{reshape}(\text{MLP}_{\mathcal{Q}^{\text{F}}}^{\text{processor}}([[\mathbf{q}_{j, \mathcal{F}_s \to \mathcal{F}_r}, \mathbf{v}_{s_j}, \mathbf{v}_{r_j}]_{j=1,2,3}])) \tag{4}$$

This does not update the three latent vectors independently, but rather all nine input latent vectors $[[\mathbf{q}_{j, \mathcal{F}_s \to \mathcal{F}_r}, \mathbf{v}_{s_j}, \mathbf{v}_{r_j}]_{j=1,2,3}]$ contribute to all updated face-face edge latent vectors $Q_{\mathcal{F}_s \to \mathcal{F}_r}'$.

Finally, we perform the node update based on the incoming messages from face-face edges, as well as other edges between the individual nodes:

$$\mathbf{v}'_i = \mathrm{MLP}_{\mathcal{V}}^{\mathrm{processor}}\big(\big[\mathbf{v}_i, \sum_{\forall e_{v_s \to v_r}/v_r = v_i} \mathbf{e}_{v_s \to v_r}', \sum_{\forall q_{\mathcal{F}_s \to \mathcal{F}_r}/\mathcal{F}_r[j] = v_i} \mathbf{q}'_{j, \mathcal{F}_s \to \mathcal{F}_r}\big]\big) \qquad (5)$$

The second term corresponds to standard edge aggregation(Battaglia et al., 2018), and the third term sums over the set of face-face edges for which $v_i$ is a receiver. Crucially, $\mathbf{q}'_{j, \mathcal{F}_s \to \mathcal{F}_r}$ selects the specific part of the face-face edge latent corresponding to $v_i$ as a receiver, i.e. $Q'_{\mathcal{F}_s \to \mathcal{F}_r}$, selects the first, second, or third vector, depending on whether $v_i$, is the first, second or third vector.

### 3.1.3 DECODER AND POSITION UPDATER

Following the approach of MeshGraphNets (Pfaff et al., 2021), the model predicts the acceleration $\mathbf{a}_i$ by applying an MLP to the final mesh node latents $\mathbf{v}_i$, which adds the bias for inertial dynamics to the model predictions. Finally, we update the node positions through a second order Euler integrator: $\mathbf{x}_i^{t+1} = \mathbf{a}_i + 2\mathbf{x}_i^t - \mathbf{x}_i^{t-1}$.

### 3.1.4 IMPLEMENTATION

**Object-level nodes**    In a perfect rigid, information propagates across the rigid at infinite speed. However, with $n$ message passing steps, information can only propagate $n$ hops, and for a very fine mesh when a collision happens it may take *many* message passing steps to reach all other nodes in the mesh. To facilitate long range communication, we add a virtual object-level node $\mathbf{v}^O$ at the center of each rigid object, and connect it to all of the mesh nodes of the object via bidirectional mesh-object edges. Ultimately, these extra edges and nodes will also participate in message passing equations 3 and 5, as described in the Appendix. This design allows completely decoupling the dynamics from mesh complexity as collision events can be computed on local neighborhoods using local message passing on edges $\mathcal{E}$, and the resulting collision impulses can be aggregated and broadcasted using the virtual object node $\mathbf{v}^O$. Note that unlike (Mrowca et al., 2018; Li et al., 2019a), we use only a single object node which itself only communicates with nodes on the surface of the object.

**Loss and model training**    Following MeshGraphNets (Pfaff et al., 2021), we define the loss as the per-node mean-squared-error on the predicted accelerations $\mathbf{a}_i$, whose ground truth values are estimated from the 2 previous positions and the future target position. We train the model using the Adam optimizer (Kingma & Ba, 2014). To stabilize long rollouts, we add random-walk noise to the positions during training (Sanchez-Gonzalez et al., 2020; Pfaff et al., 2021; Allen et al., 2022). We also augment the dataset by rotating the scenes by random angles $\theta$ around the $Z$ axis.

**Generating rollouts and evaluation**    At test time we generate trajectories by iteratively applying the model to previous model predictions. Following Allen et al. (2022); Müller et al. (2005), we use shape matching to preserve the shape of the object during rollouts. We use the predicted positions of the nodes in the object mesh to compute the position of the object center, as well as the rotation quaternion with respect to the reference mesh $M^U$. Then we compute the new node positions by applying this rotation to the reference mesh $M^U$ and shifting the center of the object. This is performed only during inference, not during training.

### 3.2 BASELINES

Our baselines study the importance of how we represent and resolve collisions in learned simulators. In particular, we consider the baseline outlined in Figure 3 which detect collisions in different ways by representing objects with particles, or meshes, and representing collisions between nodes or faces.

**MeshGraphNets**    (MGNs) (Pfaff et al., 2021) are similar to FIGNet in that they represent the object as a mesh, with messages propagated along the mesh surface. However, when detecting collisions between objects, they connect *nodes* that are within some fixed distance $d_c$. For objects that have sparse meshes (and may collide along an edge or face, see Figure 3), collisions may

go undetected if the edge is sufficiently long relative to $d_c$. **MeshGraphNets-LargeRadius** is an attempt to fix this issue of MGNs by increasing the radius $d_c$ to the longest edge length in the mesh to ensure collisions are detected. However, choosing such a large radius can lead to an explosion in the number of detected collisions, which significantly slows down computation (see Results). MGNs also do not use an object-level node for efficient message-passing across the object.

**DPI Reimplemented** If we relax the requirement that objects are represented as meshes, another popular way to represent objects is as collections of particles (Li et al., 2019b). Particles are typically densely sampled throughout an object's volume which allows for collision detection and efficient message propagation, but this then requires using many more nodes. We reimplement DPI (Li et al., 2019b) as a particle-based baseline by converting meshes to particle point clouds using a radius of 0.025. We use the loss formulation from DPI (see Appendix), and predict velocities rather than accelerations.

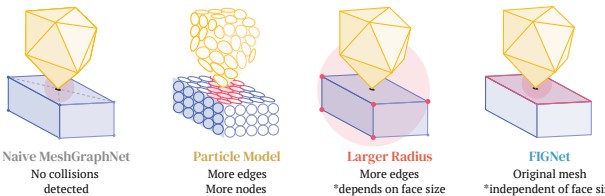

Figure 3: Schematic of how baselines treat interacting objects. If one object's vertex intersects another object's face, a small collision radius $d_c$ will not detect a collision (Naive MeshGraphNet). A dense particle representation fixes the problem but adds more edges and more nodes. Using a larger collision radius introduces quadratically more edges. By operating on faces, FIGNet can use a small collision radius without missing collisions.

**Representation of the floor** In all of our datasets, the floor is the largest and simplest object. While FIGNet has the advantage of being able to represent the floor using only two large triangles, the baselines require a denser floor to ensure that collisions with the floor are detected. In practice, this means that we refine the floor mesh so that triangle edges are not longer than 1.5 units (compared to a maximum object size of 1.5). These variants are marked with $+$ throughout the results.

For more complex datasets, refining the floor mesh can lead MGN and DPI models to run out of memory during training due to excessive numbers of nodes and collision edges. When this happens, we replace the floor mesh by an implicit floor representation (Sanchez-Gonzalez et al., 2020; Allen et al., 2022) where each node includes a feature representing the distance to the floor, clamped to be within $d_c$. These variants are marked with $*$ throughout the results. Note that this technique is fundamentally limited: it assumes that there is a fixed number of immovable large objects (the floor, or optionally also the walls), and therefore this model cannot be applied to scenes with varying numbers of walls or floor planes either during training or inference.

## 4 RESULTS

Here we investigate FIGNet's ability to model both simulated rigid body scenes as well as real world rigid interactions. We focus on investigating three key aspects of FIGNet: its efficiency, accuracy, and generalization. Across simple and complex simulated scenes, FIGNet outperforms node-based baselines in computational efficiency (by up to 8x) and in rollout accuracy (up to 4-8x). FIGNet also generalizes remarkably well across object and floor geometries far outside its training distribution. Finally, we demonstrate that FIGNet approximates real world planar pushing data better than carefully tuned analytic simulators like PyBullet (Coumans, 2015), MuJoCo (Todorov et al., 2012), and hand-crafted analytic pushing models (Lynch, 1992). Throughout the results, we compare to the representative model variations outlined in Figure 3. For rollout videos, please refer to the website: sites.google.com/view/fig-net.

**Efficient modeling for simple meshes** As alluded to previously, even relatively simple meshes can be challenging to model for node-based techniques. To accurately model the interaction between faces or edges of meshes, node-based models must increase their computational budget by either remeshing an object so the distance between nodes is smaller, or by increasing their collision radius $d_c$ to ensure that possible collisions are detected. This is particularly problematic when objects have simple meshes which mostly consist of large faces, like cubes and cylinders.

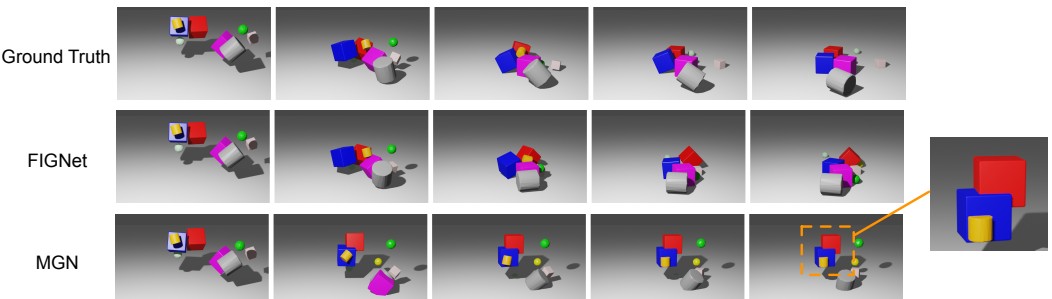

Figure 4: Rollout examples on Movi-A dataset. The node-based MeshGraphNet model struggles to resolve the collision between the sides of the sparse cube meshes.

The Kubric MOVi-A dataset (Greff et al., 2022) perfectly encompasses this scenario. In this dataset, 3-10 rigid objects (cubes, spheres or cylinders) are tossed onto a floor. The distance between nodes *within* an object's geometry can range from 0.25 to 1.4. To compare models, we measure the root mean squared error (RMSE) between the predicted and ground truth trajectories for translation and rotation after 50 rollout steps.

As expected, because the object geometries are sparse, using the node-based MGN with the same collision radius as FIGNet ($d_c$=0.1) exhibits poor prediction performance, with objects often inter-penetrating one another (Figure 4) leading to poor translation and rotation errors relative to ground truth (Figure 5a). Increasing the collision radius to 1.5 solves these problems, but at great computational cost with an explosion in the number of collision edges that must now be considered (Figure 5a). This is especially true for DPI-Nets ($d_c$=0.5), the particle-based method, which can no longer represent the floor as sets of particles without running out of memory during training. By comparison, FIGNet predictions are plausible with low translation and rotation errors (Figure 4), while not increasing the number of collision edges as dramatically, maintaining efficiency.

**Modeling unprecedentedly complex scenes** FIGNet's representation of object geometries as faces enables more accurate predictions for more complex meshes without sacrificing efficiency. The Kubric MOVi-B dataset (Greff et al., 2022) contains much more complex shapes, including teapots, gears, and torus knots, with a few hundred up to just over one thousand vertices per object.

Despite the incredible complexity of this dataset, FIGNet produces exceptionally good rollouts that closely track the ground truth dynamics (Figure G.3). FIGNet has 4x lower translation RMSE, and 2x lower rotation rotation RMSE compared to other models (Figure 5b). Note that the meshes in MOVi-B are complex enough that all baselines run out of memory if applied to MOVi-B directly with a densely sampled floor. Therefore, all baselines use the implicit floor distance feature, which greatly reduces the number of overall edges (since there will be no de-

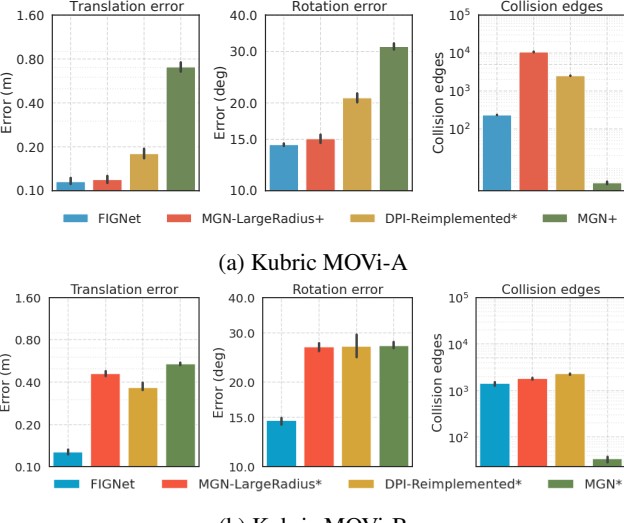

(a) Kubric MOVi-A

(b) Kubric MOVi-B

Figure 5: Performance for FIGNet and several baseline methods: root mean squared-error on translation and rotation after 50 timesteps, and number of collision edges for (a) MOVi-A and (b) MOVi-B datasets. FIGNet has significantly better translation and rotation error, while also creating fewer collision edges overall. Models with * use an implicit floor representation, while models with + use a subdivided floor.

tected collisions with the floor). Even with this adjustment, the node-based baselines (DPI-Reimplemented* ($d_c$=0.25) and MGN-LargeRadius ($d_c$=0.3)) still generally require more collision

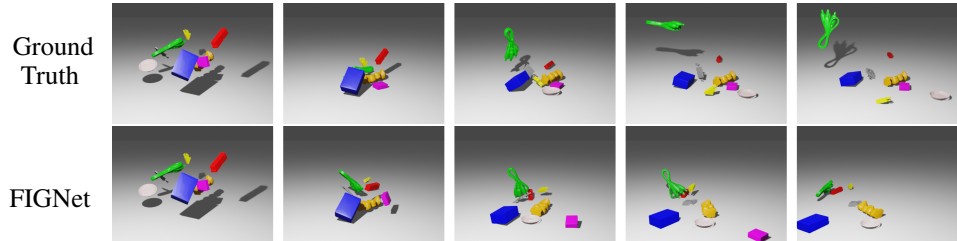

Figure 6: Rollout example on Kubric MoviC dataset (Greff et al., 2022) from a FIGNet model trained on the MOVi-B dataset. FIGNet is able to generalize to the complex geometries of the MoviC objects which it has never seen during training.

edges than FIGNet (Figure 5b). Moreover, DPI-Reimplemented* requires dense particle representations of the objects (on average ~4000 nodes per simulation compared to the ~1500 nodes in the original mesh), increasing the model memory requirements.

The performance of FIGNet here is partly driven by the object-level nodes that FIGNet uses to more efficiently propagate information through object meshes, but not entirely (see ablations in Figure E.1 and Figure E.1). This partly explains why DPI-Reimplemented* performs better than MGNs in this dataset, as it also uses object-level virtual nodes.

**Generalization to different object and floor geometries**    Given FIGNet's strong performance on MOVi-A and MOVi-B, we next sought to understand how well FIGNet generalizes across different object shapes. We take the FIGNet models pre-trained on the MOVi-A and MOVi-B datasets and apply them to the MOVi-B and MoviC respectively. In both the MOVi-A-to-MOVi-B and MOVi-B-to-MoviC generalization settings, FIGNet has 2-4x lower translation and rotation error than the baselines (Figure G.1a, Figure G.1b) and produces realistic-looking rollouts (Figure 6, Figure G.4). This is particularly remarkable in the MOVi-A-to-MOVi-B setting, because the model has only observed 3 different object shapes during training (cubes, cylinders and spheres). Furthermore, the MOVi-B translation error of FIGNet trained on MOVi-A (Figure G.1a) is *better* than any baseline models *trained directly on the MOVi-B dataset* (Figure 5b), once again demonstrating the generalization ability of face-face collisions. In contrast, DPI-Reimplemented* and both versions of MGNs struggle to generalize to scenes with new objects (Figure G.1a and Figure G.1b).

**Modeling real world dynamics better than analytic simulators**    In the preceeding sections, we saw that FIGNet significantly outperforms alternative learned models for complex, simulated datasets. However, the real world is more complicated than simulation environments – even simple dynamics like planar pushing under friction are hard to model because of sub-scale variations in object surfaces. Here we investigate how FIGNet compares to analytic simulators in modeling real-world pushing dynamics.

For these experiments, we use the MIT Pushing Dataset ((Yu et al., 2016); Figure 7a), which consists of 6000 real-world pushes for 11 different objects along 4 different surfaces (see subsection B.2

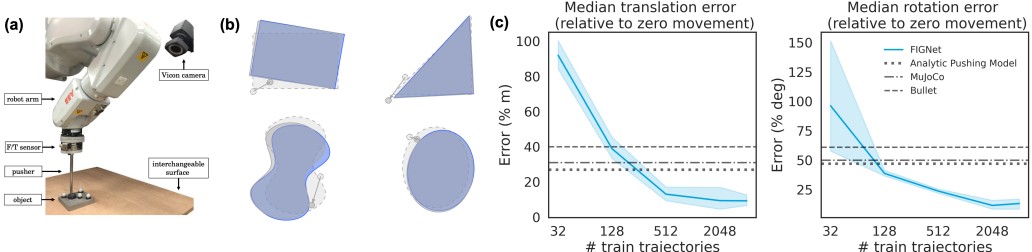

Figure 7: MIT Pushing dataset results. (a) A demonstration of the setup from Yu et al. (2016). (b) Example rollouts from FIGNet (blue) vs. ground truth (gray). The dashed outline indicates the initial object position. (c) Performance on MIT Pushing dataset for FIGNet as compared to analytic simulators MuJoCo (Todorov et al., 2012), PyBullet (Coumans, 2015), and the planar pushing analytic model developed by (Lynch, 1992) and implemented for this dataset by Kloss et al. (2022).

for details). The objects are tracked for the full push. We sample different training dataset sizes to examine FIGNet's sample efficiency and test on a set of 1000 trajectories across objects. To mitigate the effect of outlier trajectories, we report the median translation and rotation error between the ground truth and predicted trajectories after 0.5 seconds, which corresponds to 125 timesteps. To provide an interpretable unit for the metrics, we scale the rotation and translation error by the error that would be predicted by a model that always outputs zeroes. 100% error therefore corresponds to performing no better than a model which always predicts 0.

We compare FIGNet to two widely used analytic rigid body simulators, MuJoCo (Todorov et al., 2012) and PyBullet (Coumans, 2015), as well as the planar pushing specific model from Lynch (1992). For tuning system parameters of the analytic simulators, we use Bayesian Optimization on the training dataset with all available friction parameters, as well as the "timestep" for MuJoCo and PyBullet. Futhermore, while MuJoCo and PyBullet trajectories are evaluated based on the tracked object positions directly (and are therefore prone to perception errors), the planar pushing analytic model uses the metadata for each pushing trajectory (giving information about the pushing direction, velocity, etc. without perception noise). This allows us to separately examine whether FIGNet improves on analytic simulators because it can compensate for perception errors (where it would perform better than PyBullet and MuJoCo only) or because it can model pushing under friction more effectively (where it would perform better than all analytic models). We evaluate the analytic models on the same test dataset as FIGNet.

Remarkably, FIGNet outperforms all analytic simulators trained on only 128 trajectories of real world data (Figure 7c). Unlike MuJoCo and PyBullet, FIGNet is robust to errors in perception which occasionally reveal the pusher moving an object even when not in contact, or when the pusher penetrates the object. However, because FIGNet also improves on the analytic simulator (Lynch, 1992), this means that it is also simply modeling pushing under friction more accurately than can be analytically modeled. Its particular improvement in rotation error relative to the analytic models suggests that it can model torsional friction more accurately. This is exciting as it opens new avenues for considering when learned models may be preferable to more commonly used analytic simulators.

## 5 DISCUSSION

We introduced FIGNet, which models collisions using multi-indexed face-to-face interactions, and showed it can capture the complex dynamics of multiple rigid bodies colliding with one another. Compared to prior node-based GNN simulators, FIGNet can simulate much more complex rigid body systems accurately and efficiently. It shows strong generalization properties both across object shapes, and even to different floor geometries. On real-world data, FIGNet outperforms analytic simulators in predicting the motion of pushed objects, even with relatively little data, presenting an exciting avenue for future development of learned simulators that model real world behavior better than current analytical options, such as MuJoCo (Todorov et al., 2012) or PyBullet (Coumans, 2015).

Our face-to-face approach is not limited to triangular faces, nor rigid body interactions. Extensions to quadric, tetrahedral, etc. faces can use the same message-passing innovations we introduced here. Similarly, while this work's focus was on rigid body dynamics, in principle FIGNet naturally extends to multi-material interactions. Soft bodies or fluids can be represented by collections of particles, and their interactions with faces can be modeled similarly to FIGNet. Exciting directions for future work could investigate whether different kinds of face interactions could even be learned from different datasets, and then composed to handle few-shot dynamics generalization.

In terms of limitations, FIGNet, like other GNN-based simulators, is trained with a deterministic loss function. In practice, this works well for the complex rigid systems studied here, however for highly stochastic or chaotic systems this may not be as effective. The datasets studied here include complex geometry, however the extent to which FIGNet will scale to extremely complex meshes (millions of faces) is unclear, and other approaches, e.g., implicit models, may be more suitable. While FIGNet works better than analytic simulators for real world data, it still requires access to state information, and future work should investigate how it can be coupled with perception to enable learning from raw pixel observations.

FIGNet represents an important advance in the ability to model complex, rigid dynamics. By modeling mesh geometries directly, FIGNet avoids the pitfalls of previous node-based learned simulators.

FIGNet opens new avenues for learned simulators to be used as potential alternatives to existing analytic methods even for very complex scenes, with valuable implications for robotics, mechanical design, graphics, and beyond.

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

## A    Appendix

## B    Dataset details

### B.1    Kubric

Kubric dataset (Greff et al., 2022) consists of the rigid-body simulations of diverse 3D objects tossed simultaneously onto a large plane. The simulations are created using the Bullet simulator (Coumans, 2015). Kubric provides several datasets with increasing complexity of the object meshes: MOVi-A, MOVi-B, MoviC, etc. In this work, we train the models on MOVi-A and MOVi-B, as those already present a challenge to baseline models. We also provide the generalization experiments on MoviC.

Each trajectory in Kubric MOVi-A contains 3-10 objects of different sizes tossed toward the given position on the floor. In MOVi-A, only simple shapes are used: cubes (51 nodes), spheres (64 nodes), and cylinders (64 nodes). MOVi-B and MoviC contain simulations with 11 and 1030 objects respectively (the latter taken from the Google Scanned Objects (Downs et al., 2022)), with between 51 and several thousand nodes.

The objects have two types of material properties: metal (friction 0.4, restitution 0.3, density 2.7) or rubber (friction 0.8, restitution 0.7, density 1.1) for MOVi-A or MOVi-B, For MoviC, the material parameters are the same for all objects (friction 0.5, restitution 0.5 and density 1.0). We append mass, friction and restitution to the features for each node of the object.

The Kubric dataset was originally created for perception tasks and includes images of the generated trajectories. In this work, we use only the information about the physical state, consisting of the rest position of the mesh, the object position and rotation quaterion. To obtain the state information, we re-generate the dataset using the github code `github.com/google-research/kubric`. For each of MOVi-A, MOVi-B and MoviC datasets, we use 1500 trajectories for training, 100 for validation and 100 for testing. Each trajectory consists of 96 time steps.

### B.2    MIT Pushing

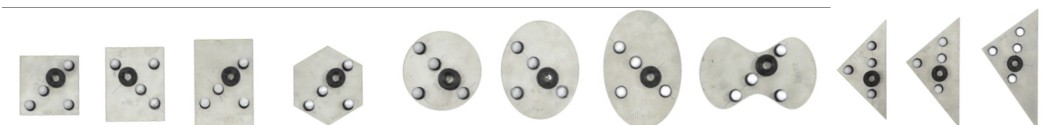

Figure B.1: Objects used in the MIT pushing dataset. Figure credit to Yu et al. (2016).

The MIT Pushing Dataset introduced by Yu et al. (2016) consists of 6000 real-world pushes for 11 different objects (Figure B.1) along 4 different surfaces, resulting in a total of 264000 trajectories. The trajectories vary in that the pusher's velocity and acceleration can change, as well as the point of contact between the pusher and the object, and the angle between the pusher and the surface of the object where it makes contact. Object poses and pusher poses were recorded at a rate of 250 Hz, with each push unfolding over 5cm. We preprocess the data using the script provided by Kloss et al. (2022) at `github.com/mcubelab/pdproc`.

Trajectories from this dataset consist of the positions and rotations of the object and robotic pushing tip. To compare results to Kloss et al. (2022), we take the subset of trajectories which have a pusher acceleration of 0. Each trajectory is cut off after 0.5 seconds have passed, corresponding to 125 timesteps. From the full set of trajectories, we randomly choose 10000 trajectories across all objects from the "abs" surface material for training, and 1000 for testing. Of these 10000 trajectories, we sample different dataset sizes for training to examine FIGNet's sample efficiency.

For this dataset, we do not use static properties of objects in the node features, other than to indicate that nodes with mass 0 belong to the controlled pusher. However, we do subsample trajectories for training and rollouts to once every 3 timesteps, which stabilizes training. To compensate for errors in state estimation in the dataset, we found a larger node noise value was also critical (0.001). Because this dataset has a controlled pusher, we also do not predict the pusher's position. We predict only the

movement of the object mesh. Otherwise, all training details and architecture choices were identical to those used for the Kubric dataset.

## C    METHOD DETAILS

### C.1    FULL MODEL DESCRIPTION

FIGNet uses an Encode-Process-Decode approach similar to (Sanchez-Gonzalez et al., 2020; Pfaff et al., 2021). Crucially the encoder and processor are adapted to be able to support multiple node and edge types simultaneously. Also we introduce a new type of edge update function for message passing through face-face edges, which each have three sender and three receiver nodes, rather than just one. Note in our case face-face edges are always triangle-triangle edges, hence the three senders and receivers, but this approach would generalize to any other type of edges, such as triangle-point edges, tetrahedron-triangle edges, segment-triangle edges, etc. Also note that our approach is a strict generalization of message passing between pair of nodes, and when applied to point-point edges, it recovers all aspects of regular edge updates.

#### C.1.1    ENCODER

The encoder constructs the input graph $\mathcal{G} = (\mathcal{V}^{\mathrm{M}}, \mathcal{V}^{\mathrm{O}}, \mathcal{E}^{\mathrm{M}}, \mathcal{E}^{\mathrm{OM}}, \mathcal{E}^{\mathrm{MO}}, \mathcal{Q}^{\mathrm{F}})$ that represents the scene from the mesh $M^t$ with 2 different node types, 3 different regular edge types, and 1 face-face edge type. Each node type has some features associated with it, which are encoded into fixed-size latent spaces.

**Mesh nodes**    $\mathcal{V}^{\mathrm{M}}$ represents the set containing each of the mesh nodes $v_i^{\mathrm{M}}$, with input features $\mathbf{v}_i^{\mathrm{M,features}} = [\mathbf{x}_i^t - \mathbf{x}_i^{t-1}, ..., \mathbf{x}_i^{t-h+1} - \mathbf{x}_i^{t-h}, \mathbf{p}_i, k_i, \mathbf{f}_i^t]$, where $\mathbf{x}_i^t$ is the position of the node at time $t$, $\mathbf{p}_i$ are static object properties, $k_i$ is a binary "kinematic" feature that indicates whether the node is subject to dynamics (e.g. the moving objects), or its position is set externally (e.g. the floor), and $\mathbf{f}_i^t = k_i(\mathbf{x}_i^{t+1} - \mathbf{x}_i^t)$ is a feature that indicates how much kinematic nodes are going to move at the next time step. We use an MLP to encode all of these features into an initial latent representation for mesh nodes:

$$\mathbf{v}_i^{\mathrm{M}} = \mathrm{MLP}_{\mathcal{V}^{\mathrm{M}}}^{\mathrm{encoder}}(\mathbf{v}_i^{\mathrm{M,features}})$$

**Object nodes**    $\mathcal{V}^{\mathrm{O}}$ represents the set containing each of the object nodes representing each rigid body $v_i^{\mathrm{O}}$, with input features $\mathbf{v}_i^{\mathrm{O,input}}$ analogous to those of the mesh nodes, using the center of mass of the object as the position for the object node. We use an MLP to encode all of these features into an initial latent representation for mesh nodes:

$$\mathbf{v}_i^{\mathrm{O}} = \mathrm{MLP}_{\mathcal{V}^{\mathrm{O}}}^{\mathrm{encoder}}(\mathbf{v}_i^{\mathrm{O,input}})$$

**Mesh edges**    $\mathcal{E}^{\mathrm{M}}$ are bidirectional edges added between mesh nodes that are connected in the mesh. For each mesh edge $e_{v_{\mathrm{s}}^{\mathrm{M}} \to v_{\mathrm{r}}^{\mathrm{M}}}^{\mathrm{M}}$ connecting a sender mesh node $v_{\mathrm{s}}^{\mathrm{M}}$ to a receiver mesh node $v_{\mathrm{r}}^{\mathrm{M}}$ we build edge features $\mathbf{e}_{v_{\mathrm{s}}^{\mathrm{M}} \to v_{\mathrm{r}}^{\mathrm{M}}}^{\mathrm{M,features}} = [\mathbf{d}_{\mathrm{rs}}, \mathbf{d}_{\mathrm{rs}}^U]$, where $\mathbf{d}_{\mathrm{rs}} = v_{\mathrm{r}}^{\mathrm{M}} - v_{\mathrm{s}}^{\mathrm{M}}$ is a vector of position differences between the nodes in the currently rotated mesh $M^t$ (e.g. the positions of the nodes at the current of timestep, which are also affected by training noise); and $\mathbf{d}_{\mathrm{rs}}^U$ is the position difference in the reference mesh $M^U$ (which is static and independent of the dynamics). We use an MLP to encode all of these features into an initial latent representation for mesh nodes:

$$\mathbf{e}_{v_{\mathrm{s}}^{\mathrm{M}} \to v_{\mathrm{r}}^{\mathrm{M}}}^{\mathrm{M}} = \mathrm{MLP}_{\mathcal{E}^{\mathrm{M}}}^{\mathrm{encoder}}(\mathbf{e}_{v_{\mathrm{s}}^{\mathrm{M}} \to v_{\mathrm{r}}^{\mathrm{M}}}^{\mathrm{M,features}})$$

**Object-Mesh edges**    $\mathcal{E}^{\mathrm{OM}}$ are unidirectional edges that go from each of the object nodes, to all of the mesh nodes that belong to that object. There is one object-mesh edge $e_{v_{\mathrm{s}}^{\mathrm{O}} \to v_{\mathrm{r}}^{\mathrm{M}}}^{\mathrm{OM}}$ for each mesh node in $\mathcal{V}^{\mathrm{M}}$. Input features $\mathbf{e}_{v_{\mathrm{s}}^{\mathrm{O}} \to v_{\mathrm{r}}^{\mathrm{M}}}^{\mathrm{OM,features}}$ are analogous to the mesh edges, computed from the positions of the object and mesh nodes. We use an MLP to encode all of these features into an initial latent representation for mesh nodes:

$$\mathbf{e}_{v_{\mathrm{s}}^{\mathrm{O}} \to v_{\mathrm{r}}^{\mathrm{M}}}^{\mathrm{OM}} = \mathrm{MLP}_{\mathcal{E}^{\mathrm{OM}}}^{\mathrm{encoder}}(\mathbf{e}_{v_{\mathrm{s}}^{\mathrm{O}} \to v_{\mathrm{r}}^{\mathrm{M}}}^{\mathrm{OM,features}})$$

**Mesh-Object edges** $\mathcal{E}^{\mathrm{MO}}$ are unidirectional edges that go from each of the mesh nodes, to the object node representing the object it belongs to. There is also one mesh-object edge $e^{\mathrm{MO}}_{v^{\mathrm{M}}_{\mathrm{s}} \to v^{\mathrm{O}}_{\mathrm{r}}}$ for each mesh node in $\mathcal{V}^{\mathrm{M}}$. Input features $\mathbf{e}^{\mathrm{MO,features}}_{v^{\mathrm{M}}_{\mathrm{s}} \to v^{\mathrm{O}}_{\mathrm{r}}}$ are analogous to the mesh edges, computed from the positions of the mesh and object nodes. We use an MLP to encode all of these features into an initial latent representation for mesh nodes:

$$\mathbf{e}^{\mathrm{MO}}_{v^{\mathrm{M}}_{\mathrm{s}} \to v^{\mathrm{O}}_{\mathrm{r}}} = \mathrm{MLP}^{\mathrm{encoder}}_{\mathcal{E}^{\mathrm{MO}}}(\mathbf{e}^{\mathrm{MO,features}}_{v^{\mathrm{M}}_{\mathrm{s}} \to v^{\mathrm{O}}_{\mathrm{r}}})$$

**Face-face edges** $\mathcal{Q}^{\mathrm{F}}$ are edges that connect two mesh faces belonging to different objects. A face-face edge $q^{\mathrm{F}}_{\mathcal{F}_s \to \mathcal{F}_r}$ is added if any point of the sender face $\mathcal{F}^{\mathrm{M}}_s$ is within the collision radius $d_c$ from any point of the receiver face $\mathcal{F}^{\mathrm{M}}_r$. This is a new type of edge, because from the point of view of the nodes of the graph, this is a "directed hyper edge", that connects the three mesh nodes of the sender mesh face $\mathcal{F}^{\mathrm{M}}_s = (v^{\mathrm{M}}_{\mathrm{s}1}, v^{\mathrm{M}}_{\mathrm{s}2}, v^{\mathrm{M}}_{\mathrm{s}3})$, to the three mesh nodes of the receiver mesh face $\mathcal{F}^{\mathrm{M}}_s = (v^{\mathrm{M}}_{\mathrm{r}1}, v^{\mathrm{M}}_{\mathrm{r}2}, v^{\mathrm{M}}_{\mathrm{r}3})$.

For each face-face edge, we build features, $\mathbf{q}^{\mathrm{F,input}}_{\mathcal{F}^{\mathrm{M}}_s \to \mathcal{F}^{\mathrm{M}}_r} = [\mathbf{d}_{\mathrm{rs}}, [\mathbf{d}_{s_j}]_{j=1,2,3}, [\mathbf{d}_{r_j}]_{j=1,2,3}, \mathbf{n}_{\mathrm{r}}, \mathbf{n}_{\mathrm{s}}]$. The features are defined with respect to the "closest collision points" between the two faces $p_{\mathrm{s}}$ (on the sender face $\mathcal{F}_s$), and $p_{\mathrm{s}}$ (on the receiver face $\mathcal{F}_r$). Geometrically, the closest point in the face might be either inside of the face, on one of the triangle edges or at one of the nodes. Then the features are defined as follows: (1) the relative vector between the closest points $\mathbf{d}_{\mathrm{rs}} = \mathbf{p}_{\mathrm{r}} - \mathbf{p}_{\mathrm{s}}$ for the two faces, (2) the spanning vectors of three nodes of the sender face $\mathcal{F}_s$ relative to the closest point at that face $\mathbf{d}_{s_i} = \mathbf{x}_{s_i} - \mathbf{p}_{\mathrm{s}}$, (3) the spanning vectors of three nodes of the receiver face $\mathcal{F}_r$ relative to the closest point at that face $\mathbf{d}_{r_i} = \mathbf{x}_{r_i} - \mathbf{p}_{\mathrm{r}}$, and (4) the face normal unit-vector of the sender and receiver faces $\mathbf{n}_{\mathrm{s}}$ and $\mathbf{n}_{\mathrm{r}}$, pointing towards the outside of the object.

We use an MLP, followed by a reshape operation, to encode all of these features into an initial latent representation for each face-face edge as **three** vectors, one for each receiver node: $\mathbf{q}^{\mathrm{features}}_{\mathcal{F}_s \to \mathcal{F}_r}$ into three face-face edge latent vectors, one for each receiver node $\mathcal{F}^{\mathrm{M}}_s = (v^{\mathrm{M}}_{\mathrm{r}1}, v^{\mathrm{M}}_{\mathrm{r}2}, v^{\mathrm{M}}_{\mathrm{r}3})$ of each face-face interaction:

$$Q_{\mathcal{F}^{\mathrm{M}}_s \to \mathcal{F}^{\mathrm{M}}_r} = [\mathbf{q}_{j,\mathcal{F}_s \to \mathcal{F}_r}]_{j=1,2,3} = \mathrm{reshape}(\mathrm{MLP}^{\mathrm{encoder}}_{\mathcal{Q}^{\mathrm{F}}}(\mathbf{q}^{\mathrm{features}}_{\mathcal{F}^{\mathrm{M}}_s \to \mathcal{F}^{\mathrm{M}}_r}))$$

Having one edge vector per edge receiver node will be a crucial aspect of the face-face edge update and mesh node update. Crucially, for each edge, and before computing features (2) and (3), we sort the nodes of the sender face $\mathcal{F}^{\mathrm{M}}_s = (v^{\mathrm{M}}_{\mathrm{s}1}, v^{\mathrm{M}}_{\mathrm{s}2}, v^{\mathrm{M}}_{\mathrm{s}3})$ and receiver face $\mathcal{F}^{\mathrm{M}}_s = (v^{\mathrm{M}}_{\mathrm{r}1}, v^{\mathrm{M}}_{\mathrm{r}2}, v^{\mathrm{M}}_{\mathrm{r}3})$ as function of the distance to the closest collision point in the corresponding face. This achieves permutation equivariance of the entire model, w.r.t. to the order in which the sender, and receiver nodes of each face are specified.

### C.1.2 MESSAGE PASSING AND ITERATIVE PROCESSOR

The goal of one step of message passing in the processor is to update all latent representations in the encoded graph based on neighborhood information. This includes mesh edge latents $\mathbf{e}^{\mathrm{M}}_{v^{\mathrm{M}}_{\mathrm{s}} \to v^{\mathrm{M}}_{\mathrm{r}}}$, object-mesh edge latents $\mathbf{e}^{\mathrm{OM}}_{v^{\mathrm{O}}_{\mathrm{s}} \to v^{\mathrm{M}}_{\mathrm{r}}}$, mesh-object edge latents $\mathbf{e}^{\mathrm{MO}}_{v^{\mathrm{M}}_{\mathrm{s}} \to v^{\mathrm{O}}_{\mathrm{r}}}$, face-face edge latents $Q_{\mathcal{F}^{\mathrm{M}}_s \to \mathcal{F}^{\mathrm{M}}_r}$, object node latents $\mathbf{v}^{\mathrm{O}}_i$, and mesh node latents $\mathbf{v}^{\mathrm{M}}_i$, to produce updated (indicated as "prime") versions of those: $\mathbf{e}^{\mathrm{M}}_{v^{\mathrm{M}}_{\mathrm{s}} \to v^{\mathrm{M}}_{\mathrm{r}}}{}'$, $\mathbf{e}^{\mathrm{OM}}_{v^{\mathrm{O}}_{\mathrm{s}} \to v^{\mathrm{M}}_{\mathrm{r}}}{}'$, $\mathbf{e}^{\mathrm{MO}}_{v^{\mathrm{M}}_{\mathrm{s}} \to v^{\mathrm{O}}_{\mathrm{r}}}{}'$, $Q_{\mathcal{F}_s \to \mathcal{F}_r}{}'$, $\mathbf{v}^{\mathrm{O}}_i{}'$ and $\mathbf{v}^{\mathrm{M}}_i{}'$. From a high level perspective the update occurs in two stages: (1) run an edge update (also referred to as "message function") on all of the different edge types (mesh edges, object-mesh edges, mesh-object edges and face-face edges), gathering information about the nodes adjacent to the edge, and (2) run a node update for each node type (mesh nodes and object nodes), aggregating information from edges of different types adjacent to the nodes.

The previous paragraph describes a single layer of message passing, but following a similar approach to Sanchez-Gonzalez et al. (2020); Pfaff et al. (2021) this layer can then be applied iteratively, as many times as necessary, with either shared or unshared neural network weights. Also similar to those we also add residual connections (sum the input to each layer to the output of the layer) at each layer, to improve gradient flow during training. Our processor consists of 10 steps of message passing, with unshared weights.

**Regular edge updates**   are used to update the mesh edge latents $\mathbf{e}^{M}_{v^{M}_{s} \to v^{M}_{r}}$, object-mesh edge latents $\mathbf{e}^{OM}_{v^{O}_{s} \to v^{M}_{r}}$, mesh-object edge latents $\mathbf{e}^{MO}_{v^{M}_{s} \to v^{O}_{r}}$. Each edge is updated following the edge update approach from (Battaglia et al., 2018):

$$\mathbf{e}'_{v^{X}_{s} \to v^{Y}_{s}} = \text{MLP}^{\text{processor}}_{\mathcal{E}^{XY}}([\mathbf{e}_{v^{X}_{s} \to v^{Y}_{s}}, \mathbf{v}^{X}_{s}, \mathbf{v}^{Y}_{r}])$$

where $X, Y$ may take the role of mesh (M) or object (O) nodes, depending on the type of edge.

**Face-face edge updates**   are used to update the face-face edge latents $Q_{\mathcal{F}^{M}_{s} \to \mathcal{F}^{M}_{r}}$. The approach to update each face-face edge is similar to regular edges, except that now each edge has three latent vectors, three senders and three receivers, and we also need to produce three output vectors:

$$Q_{\mathcal{F}_{s} \to \mathcal{F}_{r}}{}' = [\mathbf{q}'_{j,\mathcal{F}_{s} \to \mathcal{F}_{r}}]_{j=1,2,3} = \text{reshape}(\text{MLP}^{\text{processor}}_{\mathcal{Q}^{F}}([[\mathbf{q}_{j,\mathcal{F}_{s} \to \mathcal{F}_{r}}, \mathbf{v}^{M}_{s_{j}}, \mathbf{v}^{M}_{r_{j}}]_{j=1,2,3}]))$$

noting that, this does not update the three latent vectors independently, but all nine input latent vectors $[[\mathbf{q}_{j,\mathcal{F}_{s} \to \mathcal{F}_{r}}, \mathbf{v}^{M}_{s_{j}}, \mathbf{v}^{M}_{r_{j}}]_{j=1,2,3}]$ contribute to all three updated face-face edge latent vectors $Q_{\mathcal{F}_{s} \to \mathcal{F}_{r}}{}'$.

**Object node updates**   are used to update the object node latents $\mathbf{v}^{O}_{i}$. Each node is updated following the node update approach from (Battaglia et al., 2018), aggregating all of the updated edge latents (or messages) for mesh-object edges that are adjacent to that object node:

$$\mathbf{v}^{O'}_{i} = \text{MLP}^{\text{processor}}_{\mathcal{V}^{O}}\left(\left[\mathbf{v}^{O}_{i}, \sum_{\forall e^{MO}_{v^{M}_{s} \to v^{O}_{r}} / v^{O}_{r} = v^{O}_{i}} \mathbf{e}^{MO}_{v^{M}_{s} \to v^{O}_{r}}{}'\right]\right)$$

**Mesh node updates**   are used to update the mesh node latents $\mathbf{v}^{M}_{i}$. Mesh nodes may receive information from object-mesh edges, mesh edges, and face-face edges, so to update each mesh node, information of adjacent edges to that node is aggregated for each of the edge types:

$$\mathbf{v}^{M'}_{i} = \text{MLP}^{\text{processor}}_{\mathcal{V}^{M}}\left(\left[\mathbf{v}^{M}_{i}, \sum_{\forall e^{OM}_{v^{O}_{s} \to v^{M}_{r}} / v^{M}_{r} = v^{M}_{i}} \mathbf{e}^{OM}_{v^{O}_{s} \to v^{M}_{r}}{}', \sum_{\forall e^{M}_{v^{M}_{s} \to v^{M}_{r}} / v^{M}_{r} = v^{M}_{i}} \mathbf{e}^{M}_{v^{M}_{s} \to v^{M}_{r}}{}', \sum_{\forall q^{F}_{\mathcal{F}_{s} \to \mathcal{F}_{r}} / \mathcal{F}^{M}_{r}[j] = v^{M}_{i}} \mathbf{q}'_{j,\mathcal{F}_{s} \to \mathcal{F}_{r}}\right]\right)$$

where the second and third terms correspond to regular edge aggregation (just like in the object node update), and the last term sums over the set of face-face edges for which $v^{M}_{i}$ is one of the receivers and $\mathbf{q}'_{j,\mathcal{F}_{s} \to \mathcal{F}_{r}}$ selects the specific face-face edge vector corresponding to $v^{M}_{i}$ as a receiver, i.e. from $Q'_{\mathcal{F}_{s} \to \mathcal{F}_{r}}$, selects the first, second, or third vector, depending on whether $v^{M}_{i}$, is the first, second or third vector in that receiver face (recall it is always sorted by distance to the closest point in the face, so the model remains permutation equivariant).

### C.1.3   DECODER

Similar to (Sanchez-Gonzalez et al., 2020; Pfaff et al., 2021), the goal of the decoder is to produce output features. In our model we only require output features for the mesh nodes $v^{M}_{i}$. We produce this by applying an MLP decoder from the latent space of the mesh nodes after the processor, which outputs a finite-difference approximation to the acceleration:

$$\mathbf{a}_{i} = \text{MLP}^{\text{decoder}}_{\mathcal{V}^{M}}(\mathbf{v}^{M}_{i})$$

### C.2   OTHER MODEL DETAILS

**Norm features**   For all relative spatial feature vectors $\mathbf{d}$, we also also concatenated their norm $|\mathbf{d}|$ as part of the inputs.

**Training noise**   To make models stable for long rollouts while training on one-step data, we use the same strategy from (Sanchez-Gonzalez et al., 2020; Pfaff et al., 2021)to train with random walk noise in the inputs, asking the model to correct for noise in the input velocities.

**Predicted targets**   Similar to (Sanchez-Gonzalez et al., 2020; Pfaff et al., 2021) our models predicts a finite-difference acceleration that is used to update the position $\mathbf{x}_i^{t+1} = \mathbf{a}_i + 2\mathbf{x}_i^t - \mathbf{x}_i^{t-1}$.

**Normalization**   Similar to (Sanchez-Gonzalez et al., 2020; Pfaff et al., 2021) we normalized all inputs and targets to zero-mean unit-variance. The loss is computed in the normalized space of the targets. This means our entire model and loss is agnostic to the scale of the data.

**MLPs**   We use MLPs with 2 hidden layers, and 128 hidden and output sizes (except the decoder MLP, with an output size of 3). All MLPs, except for those in the decoder, are followed by a LayerNorm(Ba et al., 2016) layer.

**Optimization**   All models are trained to 1M steps with a batch size of 128 across 8 TPU devices. We use Adam optimizer, and an an exponential learning rate decay from 1e-3 to 1e-4.

## D   BASELINE DETAILS

### D.1   MUJOCO AND BULLET SIMULATORS FOR REAL-WORLD MIT PUSHING DATASET

For the MIT Pushing Dataset we compare with the MuJoCo (Todorov et al., 2012) and Bullet (Coumans, 2015) simulators. While the objects have a provided mesh geometry, the physical properties of this real-world system, as well as the optimal internal simulator parameters, are unknown and thus must be estimated by system identification.

We construct this system identification problem as a black-box optimization procedure as follows. The objective we minimize is the sum of the relative translation error and relative rotation error of the object being pushed. To measure this objective we use a 50-trajectory subset of the full training dataset; making predictions for these 50 trajectories takes 5-10 minutes for each simulator. Using more trajectories for fitting did not yield improved results. As our optimizer, we choose Bayesian optimization with Gaussian processes as implemented by Vizier (Golovin et al., 2017). We perform 500 trials of optimization, then take the best-performing hyperparameters and evaluate them on a 100-trajectory validation set.

For MuJoCo we model the externally-controlled pusher as a non-physical motion capture body attached via an equality constraint to the physically-modeled pusher geometry. This allows the simulator to impute velocities for the geometry which are unavailable when using motion capture positions directly, making the physics stabler and more realistic. We find that modeling contacts with three degrees of freedom and neglecting the differential effects of rotational and rolling friction gives strictly superior results and so perform our final optimization using this setting. We use the expensive but more accurate RK4 integrator rather than the default Euler integrator for greater precision in modeling hard contact. We search over the space of scalar friction coefficients $k \in [0, 5]$ for all three objects, as well as over the number of physics substeps $\{1, 10, 100, 1000\}$. The best-performing solution has $k_{\text{object}} = 1.91$, $k_{\text{pusher}} = 0.41$, and $k_{\text{floor}} = 0.0$, and uses 1000 physics substeps per data timestep. For Bullet we create a fixed constraint for the tip and directly update the parent location of the fixed constraint with the positions of the tip given by the input trajectories. We search over the space of lateral friction coefficients $k \in [0, 5]$ for all three objects, as well as over the number of engine substeps $\{0, 1, 2, 3, 4, 5\}$. The best-performing solution has $k_{\text{object}} = 0.28$, $k_{\text{pusher}} = 0.0$, and $k_{\text{floor}} = 1.106$, with 0 substeps.

### D.2   MESHGRAPHNETWORKS (MGN)

To make the comparison more informative and fair, we reimplement MeshNets to refer to a model with the same "bells and whisles" to ours, except for two joint ablations of important differences with (Pfaff et al., 2021).

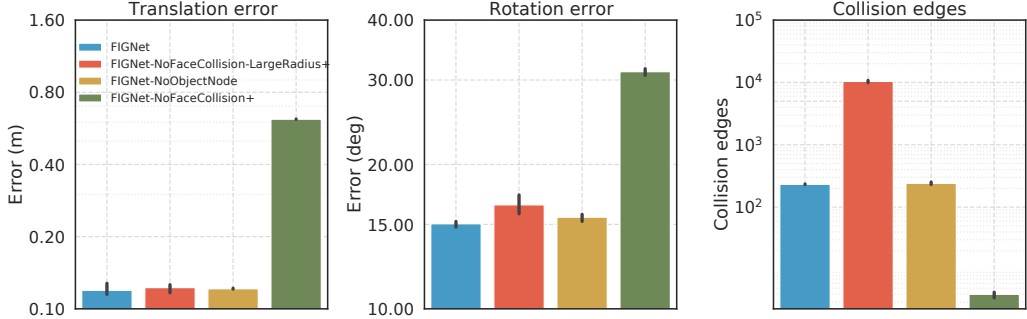

Figure E.1: Model ablations on MOVi-A dataset. Replacing the face-face collisions with the node-node collisions makes the performance deteriorate the most. Models with $^*$ use an implicit floor representation, while models with $^+$ use a subdivided floor.

**Remove object nodes (FIGNet-NoObjectNode).** The ablation consists of removing the object nodes, as well as all of the object-mesh/mesh-object edges from the graph.

**Node based collisions (FIGNet-NoFaceCollision).** The ablation consists of computing collisions according to the distance between the mesh nodes, rather than the faces, and replacing the face-face edges, by regular mesh-mesh edges, referred to as "world edges" in (Pfaff et al., 2021). Unlike MeshGraphNets, this model has the per-object node, similarly to FIGNet. In order to give this ablation a chance, we (1) use a larger collision radius $d_c$ (LargeRadius) and (2) subdivide the two gigantic triangles that make up the floor into smaller triangles of similar in size to the collision radius (1.5), or (3) when (2) becomes too we use an "implicit floor" by adding an extra feature to the nodes indicating if they are within the collision radius distance $d_c$ from the floor and by how much. We also provide some results from applying these ablations independently.

### D.3 DPI-Reimplemented

To make the comparison more informative and fair, we use DPI-Reimplemented to refer to a model with the same "bells and whisles" to ours, except for three joint ablations of important differences with (Li et al., 2019b).

**Object-level predictions (FIGNet-DPILoss)** The ablation consists of: (1) use a decoder for the object nodes, to produce a velocity/acceleration, for the position and rotation quaternion of the center of mass of the object, (2) used that prediction to update the mesh nodes positions, (3) compute the effective mesh node acceleration, and (4) use this to build the loss.

**Particles (FIGNet-Particles)** We replaced the mesh data by a dense particle representation of the object, and use those particles as nodes. As there aren't any faces we also use node-based collisions. Note in this case, we still keep the central object node, following (Li et al., 2019b) which would correspond to a $k = 1$ level hierarchy in their model.

We also provide some results from applying these ablations independently.

### E  ABLATION RESULTS

To motivate the importance of the per-object nodes and face-face collisions, we provide the ablations of our model where we remove either of these properties from the model.

Figure E.1 and Figure E.2 demonstrate the ablation on MOVi-A and MOVi-B datasets respectively. Removing the face-face collisions, but keeping the same collision radius (NoFaceCollision model) makes the performance deteriorate on both MOVi-A and MOVi-B, presumably because the current collision radius does not allow to capture all the nodes that are necessary to resolve the collision.

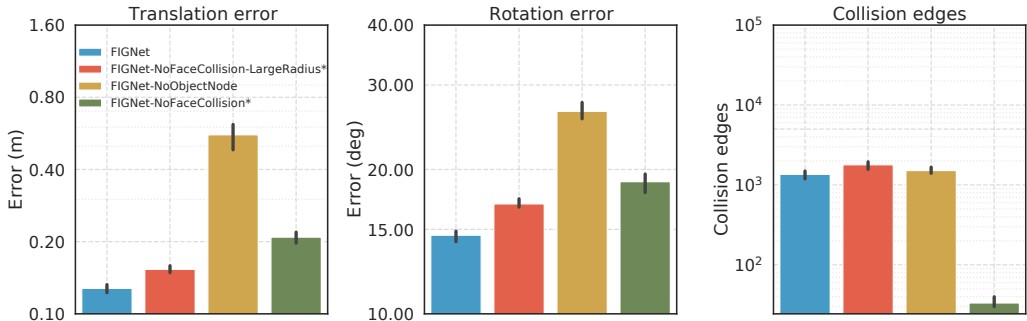

Figure E.2: Model ablations on MOVi-B dataset. Removing the object node and replacing face-face collisions with node-node drastically affects the performance, making it worse. Models with * use an implicit floor representation, while models with + use a subdivided floor.

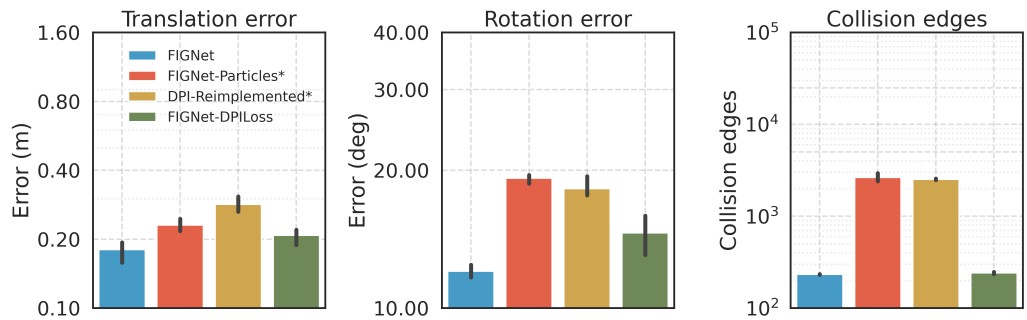

Figure E.3: Model ablations towards DPI (Li et al., 2019b) on MOVi-A dataset. While the loss formulation has little effect, replacing the mesh with densely sampled particles creates huge issues with memory requirements, and makes overall translation and rotation error worse. Models with * use an implicit floor representation.

Increasing the collision radius (NoFaceCollision-LargeRadius) restores the performance on MOVi-A, but it is not sufficient on MOVi-B to match the performance of FIGNet.

Next, we consider the ablation of removing the per-object node from FIGNet. On MOVi-A this modification does not affect the performance as much, presumably because the small number of nodes per object in MOVi-A is relatively small and it does not require additional message-passing through the object node. However, on MOVi-B we observe much higher translation and rotation errors compared to FIGNet and other baselines. This result highlights the importance of object-level node in the complex datasets like MOVi-B, as provides a "shortcut" the message-passing between the nodes of the object.

Note that the FIGNet-NoFaceCollision and FIGNet-NoFaceCollision-LargeRadius baselines replace the face-face message-passing with node-node interactions. They suffer from the same issues related to floor parameterizations as MeshNets, as desribed in the main text. Therefore we use the parameterization with subdivided floor for MOVi-A, and implicit floor for MOVi-B for these models, similarly to MeshNets (see explanation for Figure 5a and Figure 5b in the main text).

We also consider various ablations that gradually move the FIGNet model towards the DPI-Reimplemented particle-based model, as outlined in the ablations sections. Note that changing to a particle based representation makes it impossible to represent the floor using a dense representation, so all models with particles use an implicit floor (marked with * as in the main text).

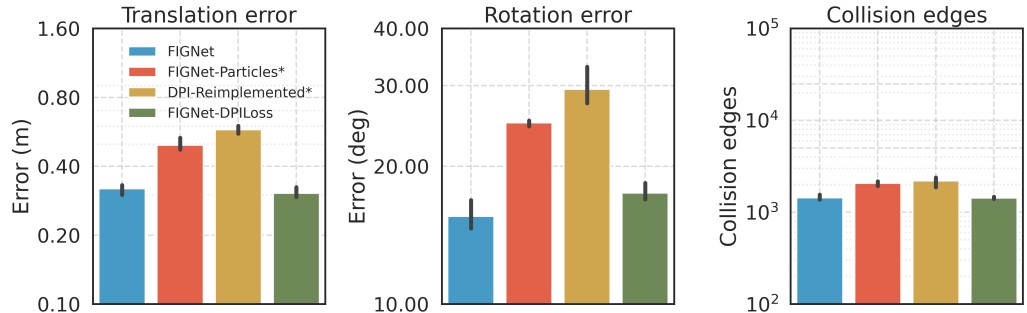

Figure E.4: Model ablations towards DPI on MOVi-B dataset. While the loss formulation has little effect, replacing the mesh with densely sampled particles creates huge issues with memory requirements, and makes overall translation and rotation error worse. Models with * use an implicit floor representation.

# F  GENERALIZATION ACROSS SURFACE GEOMETRIES

To test generalization across different surface geometries, we created custom scenes where a sphere is dropped in some location and rolls along the different surfaces (Figure F.1). Despite never seeing inclined, curved or concave planes during training, FIGNet produces plausible trajectories of how the ball should roll. This supports our findings on the remarkable ability of FIGNet to generalize to new scenes, shapes and floor landscapes.

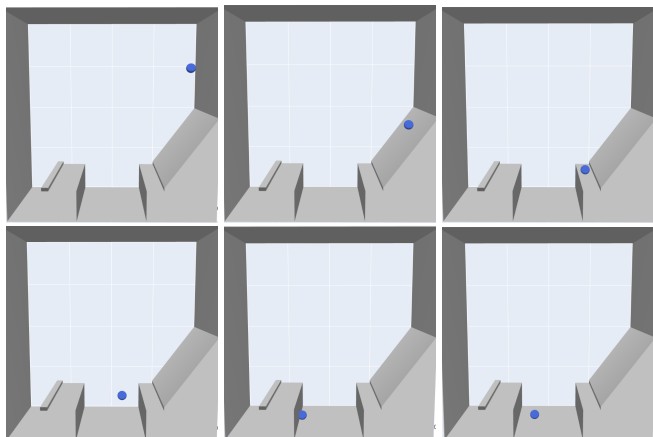

Figure F.1: Demonstration of generalization rollouts from Kubric MOVi-B to a 3D adaptation of the *Bridge* level from the Virtual Tools Game (Allen et al., 2020). Grey and blue are static and dynamic objects respectively.

# G  GENERALIZATION ACROSS OBJECT SHAPES

To test generalization across different shapes, we train models on the MOVi-A or MOVi-B datasets, and test their generalization to MOVi-B or MoviC respectively, with no further fine-tuning. In Figure G.1a and Figure G.1b, FIGNet clearly outperforms alternative methods in generalization. Even more remarkably, FIGNet's performance on MOVi-B, when trained only on MOVi-A, still surpasses all baseline performances when the baselines are trained on MOVi-B directly. This suggests that FIGNet is not only accurate when trained, but provides compelling reasons to believe it will generalize to further complex dynamics in future.

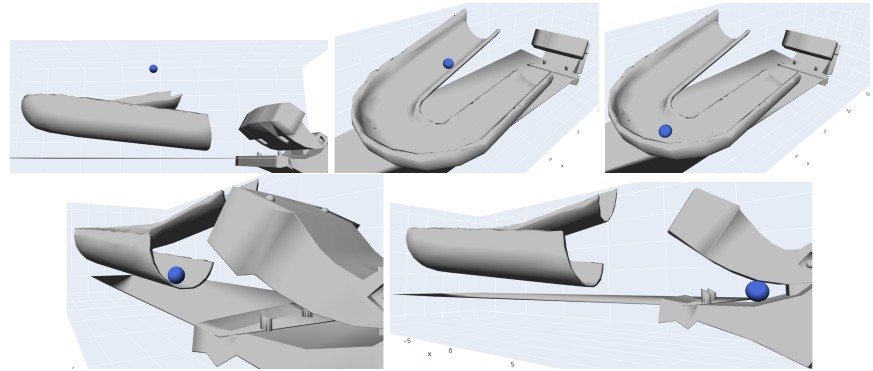

Figure F.2: Demonstration of generalization rollouts from Kubric MOVi-B to a manually designed u-slide and imported asset (`hippo.obj` asset imported from `github.com/mmacklin/tinsel.git`). Grey and blue are static and dynamic objects respectively.

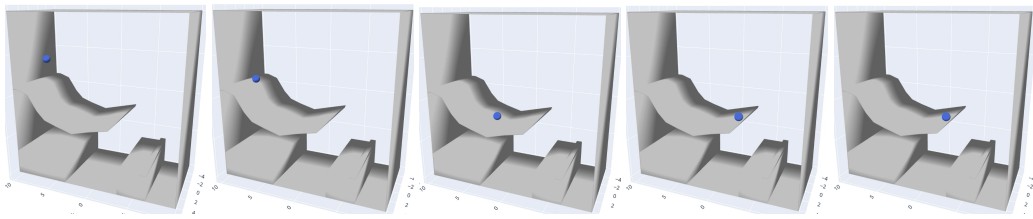

Figure F.3: Demonstration of generalization rollouts from Kubric MOVi-B to an unseen ramp.

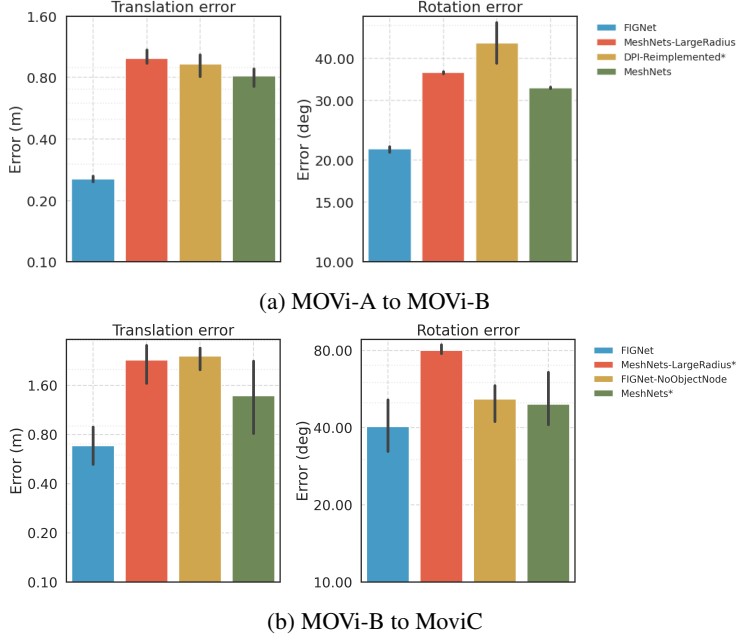

(a) MOVi-A to MOVi-B

(b) MOVi-B to MoviC

Figure G.1: Generalization performance of the FIGNet model (a) trained on Kubric MOVi-A and tested on MOVi-B (b) trained on Kubric MOVi-B and tested on MoviC.

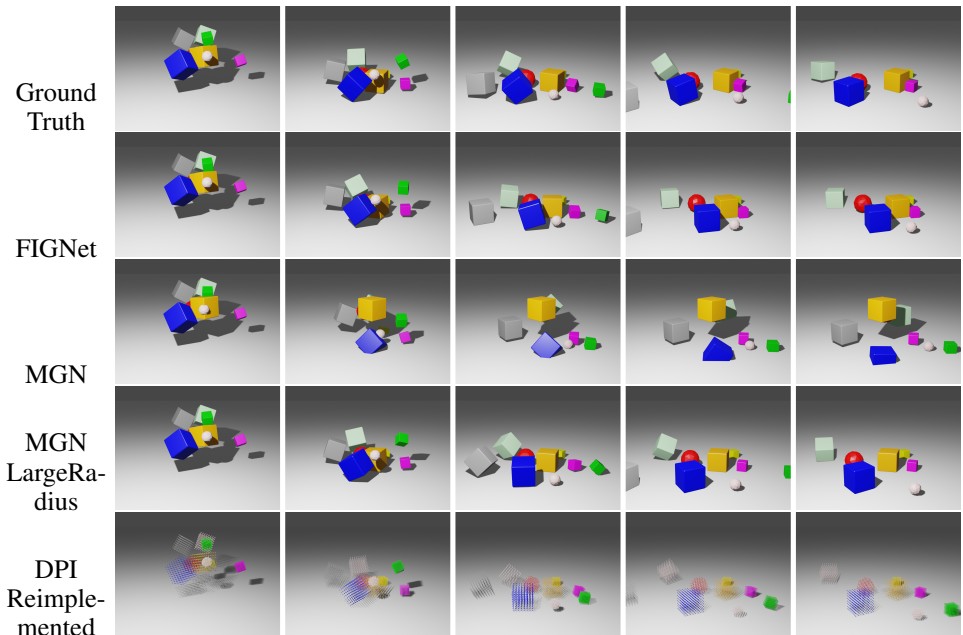

Figure G.2: Rollout examples on MOVi-A dataset

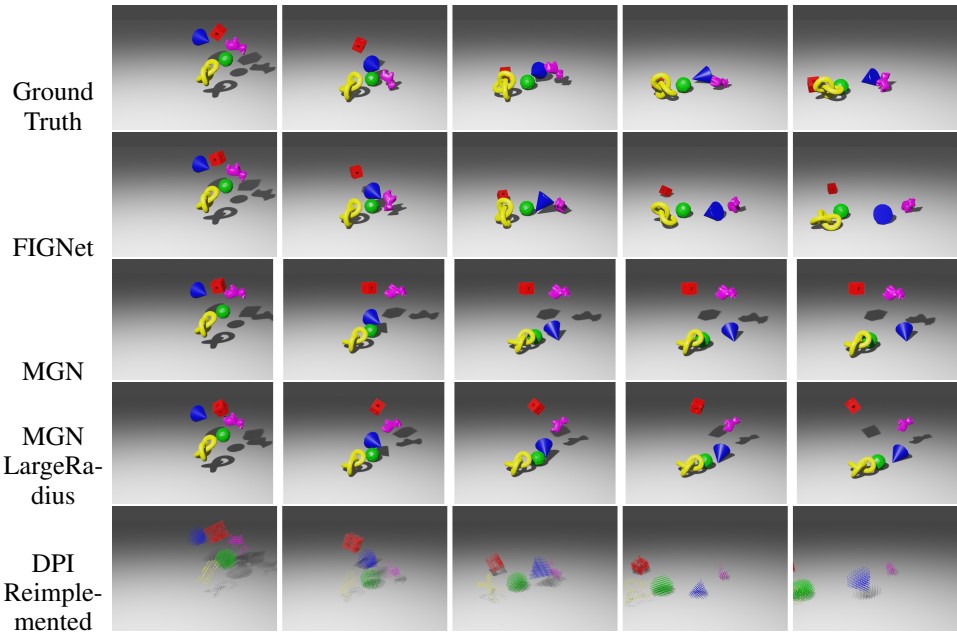

Figure G.3: Rollout examples on MOVi-B dataset

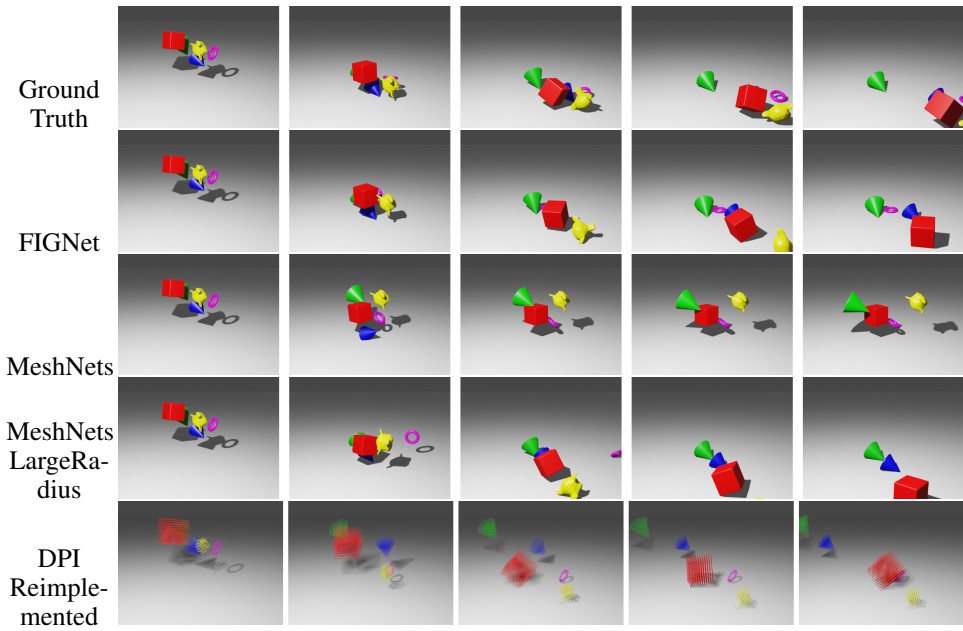

Figure G.4: Generalization from MOVi-A to MOVi-B dataset

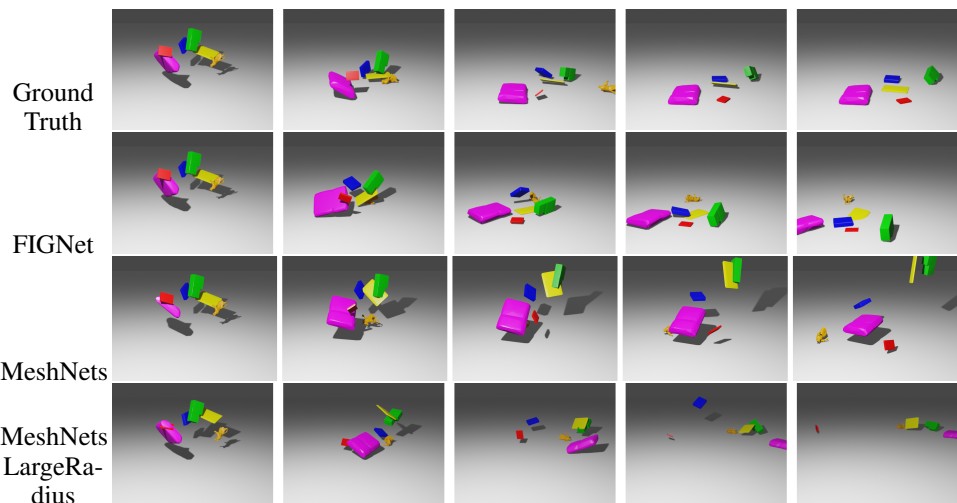

Figure G.5: Generalization from MOVi-B to Movi-C dataset

## H  Penetration measures

To measure penetration, we implement the algorithm from (Bærentzen & Aanaes, 2005). Since this algorithm is imperfect when faces are small, we report penetration statistics with respect to the ground truth trajectories given the same initial scene. FIGNet penetration distances are only 3% greater than the ground truth simulator, with only 4% more faces in penetration with each other relative to ground truth. By comparison, MeshGraphNets performs considerably worse when using a small collision radius, with penetration distances greater than 4x the ground truth model. These results are reported in Table H.1

Table H.1: Penetration distances and counts reported as a ratio to the distances and counts for the ground truth PyBullet simulator.

(a) MOVi-A dataset

| Model | Penetration distance ratio | Penetration count ratio |
|---|---|---|
| FIGNet | $1.037 \pm 0.033$ | $1.041 \pm 0.037$ |
| MGN-LargeRadius+ | $1.071 \pm 0.020$ | $1.073 \pm 0.018$ |
| MGN+ | $4.613 \pm 0.143$ | $5.246 \pm 0.187$ |

(b) MOVi-B dataset

| Model | Penetration distance ratio | Penetration count ratio |
|---|---|---|
| FIGNet | $1.121 \pm 0.034$ | $1.147 \pm 0.039$ |
| MGN-LargeRadius* | $3.813 \pm 0.985$ | $4.473 \pm 1.197$ |
| MGN* | $5.614 \pm 0.473$ | $6.703 \pm 0.573$ |

## I  Runtime

We additionally report runtime performance (on CPU) to more directly compare the amount of time taken to run a single forward step of each model. These results are reported in Table I.1 for the MOVi-A dataset and Table I.2 for the MOVi-B dataset.

FIGNet has the shortest inference runtime in comparison to baselines for the MOVi-A dataset. MGN-LargeRadius+, which is closest to FIGNet in terms of accuracy (Figure 5(a)), requires 2.7x more time to perform one inference step. DPI-Reimplemented* and MGN+ are inferior to FIGNet in terms of both runtime and accuracy.

For the MOVi-B dataset, FIGNet is slower than the alternative models, despite using fewer collision edges on average. This can be explained since the majority of the complexity for MOVi-B is due to the complexity of the mesh objects. Unlike the baselines, FIGNet contains edges connecting nodes of the mesh (mesh edges) *and* edges from the object node to each of the mesh nodes. Thus, with added object complexity, FIGNet contains more edges. Furthermore, FIGNet is the only model of the four to represent the floor explicitly, and many of the collisions that occur are between the floor and the objects. All alternative models need to represent the floor implicitly (marked with *), as otherwise they run out of memory. Future work could investigate improvements to object representations which require fewer mesh nodes to be connected, which we expect would improve FIGNet's runtime performance relative to baselines.

Table I.1: Computational performance on MOVi-A dataset (runtime and number of collision edges)

| Model | Runtime (seconds) | # Collision Edges |
|---|---|---|
| FIGNet | $0.094 \pm 0.005$ | $233 \pm 2.1$ |
| MGN-LargeRadius+ | $0.258 \pm 0.010$ | $10637 \pm 317$ |
| DPI-Reimplemented* | $0.126 \pm 0.006$ | $2515 \pm 34$ |
| MGN+ | $0.160 \pm 0.007$ | $3.8 \pm 0.39$ |

Table I.2: Computational performance on MOVi-B dataset (runtime and number of collision edges)

| Model | Runtime (seconds) | # Collision Edges |
|---|---|---|
| FIGNet | $0.342 \pm 0.012$ | $1385.610 \pm 23.818$ |
| DPI-Reimplemented* | $0.145 \pm 0.009$ | $2250.688 \pm 64.507$ |
| MGN-LargeRadius* | $0.218 \pm 0.033$ | $1797.985 \pm 59.699$ |
| MGN* | $0.175 \pm 0.018$ | $34.367 \pm 4.075$ |

## J    EFFECTS OF DIFFERENT COLLISION RADII

Figure J.1 demonstrates the ablations over different collision radii for the MeshGraphNets (MGN+) (Pfaff et al., 2021) model compared to FIGNet with collision radius 0.1 on the Kubric MOVi-A dataset. Notably, there is no collision radius where MeshGraphNets would simultaneously have comparable runtime to FIGNet while also having comparable accuracy. Large values for the collision radius lead to better accuracy, but at the cost of runtime performance. The increase in collision radius also leads to an exponential increase in collision edges in MeshGraphNets, as more nodes fall into the collision radius, leading to higher memory consumption. In our experiments, the increase in collision edges manifested as 50% increase in runtime from collision radius 0.05 to 1.5.

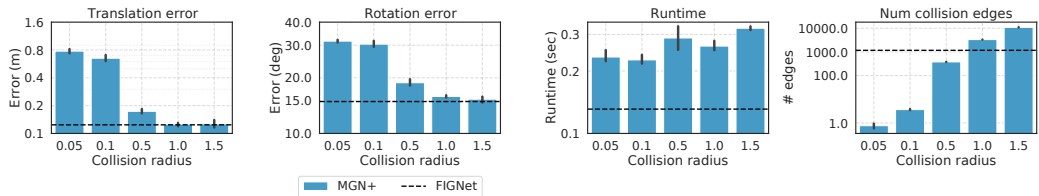

Figure J.1: Ablation over collision radii on MOVi-A dataset for MGN, compared to FIGNet with collision radius 0.1. If the nodes (in MGN) or faces (in FIGNet) between different objects lie within the collision radius, we connect them with an edge in the graph.

By comparison FIGNet is not particularly sensitive to the collision radius. We trained models with a collision radius of 0.05 and 0.5 relative to the main model's radius of 0.1. The errors for these different collision radii are given in Table J.1

Table J.1: FIGNet collision radius sensitivity analysis for MOVi-A dataset.

| Metric | Collision radius | | |
|---|---|---|---|
| | 0.05 | 0.1 | 0.5 |
| Translation error (m) | $0.151 \pm 0.014$ | $0.115 \pm 0.008$ | $0.105 \pm 0.006$ |
| Rotation error (deg) | $16.8 \pm 0.3$ | $14.4 \pm 0.2$ | $13.4 \pm 0.4$ |

# K  RESULTS FROM MAIN TEXT AS TABLES

For ease of comparison between models, we provide the results on Kubric from the main text as tables.

Table K.1: Results on Kubric datasets from main text as table

| **Kubric MOVi-A results** | | | |
|---|---|---|---|
| **Model** | **Translation error** | **Rotation error** | **Collision edges** |
| DPI-Reimplemented* | $0.180 \pm 0.017$ | $20.817 \pm 0.926$ | $2515.354 \pm 34.341$ |
| MGN-LargeRadius+ | $0.119 \pm 0.009$ | $15.069 \pm 0.649$ | $10637.350 \pm 317.336$ |
| MGN+ | $0.705 \pm 0.069$ | $31.210 \pm 0.989$ | $3.792 \pm 0.396$ |
| FIGNet | $0.115 \pm 0.008$ | $14.387 \pm 0.175$ | $232.644 \pm 2.135$ |
| **Kubric MOVi-B results** | | | |
| **Model** | **Translation error** | **Rotation error** | **Collision edges** |
| DPI-Reimplemented* | $0.368 \pm 0.057$ | $26.928 \pm 2.740$ | $2250.688 \pm 64.507$ |
| MGN-LargeRadius* | $0.460 \pm 0.045$ | $26.342 \pm 1.397$ | $1797.985 \pm 59.699$ |
| MGN* | $0.538 \pm 0.035$ | $26.914 \pm 0.783$ | $34.367 \pm 4.075$ |
| FIGNet | $0.127 \pm 0.006$ | $13.990 \pm 0.464$ | $1385.610 \pm 23.818$ |

