# OpenReview forum: "Learning rigid dynamics with face interaction graph networks"
_ICLR.cc/2023/Conference — ICLR 2023 notable top 25%_

### Official Review · Reviewer_sivF · 2022-10-22

**Confidence:** 4
**Correctness:** 4
**Technical Novelty And Significance:** 4
**Empirical Novelty And Significance:** 4
**Recommendation:** 10

**Clarity, Quality, Novelty And Reproducibility:**

The paper is very well written, the proposed model FIGNet is reasonable, interesting, novel, and seems to perform very well compared to existing baselines. The results seems reproducible, however code release would have been very much appreciated.

**Strength And Weaknesses:**

I very much appreciated this paper which introduces with great clarity a method which seems reasonable. The addition of face-level interactions in the structure of GNNs is a significant improvement over standard models, and seems to me somehow related to previous works on dual graphs, e.g. [1]. A discussion of the key differences with existing works on the subject could perhaps be added.

FIGNet seems relevant and efficient, and the design choices presented are reasonable. I only have two questions about it:

- The collision distance $d_c$ seems to be an important parameter. Did the authors measure the effect of this parameter on learning and inference? Is it a sensitive hyper-parameter? Can we imagine learning $d_c$ directly with the weights of the model?

- Sorting the nodes of a face according to the distance from the nearest point on the other face is said to be "crucial". Can the authors discuss more clearly the significance of this? Have ablation experiments been performed?

Regarding the experimental results, these are indeed "plausible". But this also seems to be the case for MeshGraphNet when the collision distance is increased. The comparison with the baselines would be easier if the results were aggregated in a table, which could be added in the appendix. The authors justify the advantage of FIGNet by evoking the exponential complexity of the baseline when $d_c$ increases. Nevertheless, FIGNet trades off the increase in the number of edges against an increase in the complexity of the model. Finally, it seems to me possible that the gain of FIGNet is quite limited in practice. Thus, a more in-depth analysis of the difference between these two models seems necessary. It would be particularly interesting to measure their inference time (and why not the number of FLOPs).

Finally, one of the weaknesses of the approach is the need to have access to a rich and detailed representation of the scene, in the form of a mesh. Have the authors considered solutions to train their models from weaker signals, or even directly from images?

In addition to the quality of the work offered, I note the use of certain tricks to keep the number of pages to the limit. In particular, the referencing of figures in the appendix in the main text is questionable, but I recognize that this is common practice. What concerns me more is the modification of the style of bibliography compared to the standards of ICLR. It seems to me that this style is mandatory (while it requires a lot of space). If this is indeed the case, I would appreciate if the authors regularize their paper, for the sake of fairness with other submissions.


**Summary Of The Paper:**

The paper introduces an extension of MeshGraphNet by adding explicit modeling of rigid body faces. The authors show that this modification significantly improves simulation performances over a large number of tasks while reducing the computational burden.

**Summary Of The Review:**

The method presented in this paper is very interesting. I particularly appreciate the clarity of the writing as well as the extensive empirical study, and the care taken to offer solid and comparable baselines. My score is however reduced by a doubt as to the practical interest of FIGNet against MeshGraphNet in the case where $d_c$ is increased, and also particularly by the liberties taken by the authors regarding the editorial rules, which is not very fair with other submissions. If the authors provide satisfactory answers to these two points, I will gladly increase my score!

## EDIT
The authors answer my remark in a very complete and convincing way. I believe that the new experiments add great value to the paper and thus highly recommend this paper for acceptance.

---

> ### Author Response · Authors · 2022-11-18
> **Response [1/2]**
>
> Thank you for the thoughtful review! We respond to your comments below.
>
> > “and seems to me somehow related to previous works on dual graphs, e.g. [1]. A discussion of the key differences with existing works on the subject could perhaps be added.
>
> [1] is not provided – could you please send us the citation so we can take a look and add it to our related work section?
>
> > The collision distance dc seems to be an important parameter. Did the authors measure the effect of this parameter on learning and inference? Is it a sensitive hyper-parameter? Can we imagine learning dc directly with the weights of the model?
>
> FIGNet is not particularly sensitive to the collision distance dc. We trained models with a collision distance of $0.05$ and $0.5$ relative to the main model's dc of $0.1$ on the MoviA dataset. The errors for these different collision distances are given in Table J.1 of the updated appendix and copied below for the reviewer’s convenience:
> Metric | dc = 0.05  | dc = 0.1  | dc = 0.5  |
> --- |--- |--- |--- |
> Translation error (m) | $ 0.151 \pm 0.014 $ | $ 0.115 \pm 0.008 $ | $ 0.105 \pm 0.006 $
> Rotation error (deg) | $ 16.817 \pm 0.339 $ | $ 14.387 \pm 0.175 $ | $ 13.440 \pm 0.411 $
>
> In its current implementation, dc is not differentiable. The collision detection procedure we use is non-differentiable, and this is the only way in which dc contributes to the loss. However, if a differentiable collision detection algorithm was created, we could use that to learn dc jointly with the physical dynamics.
>
> > Sorting the nodes of a face according to the distance from the nearest point on the other face is said to be "crucial". Can the authors discuss more clearly the significance of this? Have ablation experiments been performed?
>
> We apologize for the confusion. The word crucial was used from a theoretical perspective rather than an empirical one. In particular, in theory, without the sorting by the nearest distance, the FIGNet model is not permutation invariant. Since the three representations from the face vertices are concatenated before being passed to the network, changing the order of these vertices (which are arbitrary) would result in different feature vectors being seen by the network, possibly leading to different outputs. To make the network permutation invariant, we therefore sort the order of how the vertices are concatenated based on their distance to the collision point, which ensures that no matter how the face (triangle) is represented, the indices will always be sorted in the same way by the network. However, in practice, this turns out to not empirically matter too much (there is a very slight change in the rotation error when not using the sorting). We have removed the word “crucial” from the text, and provide the ablation below:
>
> Sorted indices by closest distance (FIGNet) vs. not sorted
> Model | Translation error (m) | Rotation error (deg)
> --- | --- | ---
> FIGNet | $ 0.115 \pm 0.008 $ | $ 14.387 \pm 0.175 $
> FIGNet without sorted indices | $0.117 \pm 0.003$ | $ 14.86 \pm 0.23 $
>
> > The comparison with the baselines would be easier if the results were aggregated in a table, which could be added in the appendix.
>
> Thank you for the suggestion, we agree! We have added the table to the appendix as Table K.1.
>
> > The authors justify the advantage of FIGNet by evoking the exponential complexity of the baseline when dc increases. Nevertheless, FIGNet trades off the increase in the number of edges against an increase in the complexity of the model. Finally, it seems to me possible that the gain of FIGNet is quite limited in practice. Thus, a more in-depth analysis of the difference between these two models seems necessary. It would be particularly interesting to measure their inference time (and why not the number of FLOPs).
>
> We agree that FIGNet is a slightly more complicated model than MeshGraphNets. However, the improvement of FIGNet over MeshGraphNets is not just run time – for the complex Movi-B dataset, its improvement in accuracy is almost 4x that of MeshGraphNets even with a large collision radius.
> We have also now measured runtime performance for the Movi-A dataset. FIGNet has the shortest inference runtime in comparison to baselines. MGN-LargeRadius+ that is closest to FIGNet in terms of accuracy (Figure 5(a)), requires 2.7x more time to perform one inference step. DPI-Reimplemented* and MGN+ are inferior to FIGNet in terms of both runtime and accuracy. Please see the table below:
>
> Model | Runtime (seconds)
> --- | ---
> FIGNet | $ 0.094 \pm 0.005  $
> MGN-LargeRadius+ | $ 0.258 \pm 0.010 $
> DPI-Reimplemented* | $ 0.126 \pm 0.006 $
> MGN+ | $ 0.160 \pm 0.007 $

---

> > ### Author Response · Authors · 2022-11-18
> > **Response [2/2]**
> >
> > > The authors justify the advantage of FIGNet by evoking the exponential complexity of the baseline when dc increases.
> >
> > To further illustrate our point about the exponential complexity of baselines, we provide an analysis of MeshGraphNets across several different collision radii on the MoviA dataset. Please see Figure J.1 of the updated appendix. Notably, there is no collision distance where MeshGraphNets would simultaneously have comparable runtime to FIGNet while also having comparable accuracy. Large values for the collision distance lead to better accuracy, but at the cost of runtime performance.
> > The increase in collision distance also leads to the exponential increase in collision edges in MeshGraphNets, as more nodes fall into the collision radius, leading to higher memory consumption. In our experiments, the increase in collision edges manifested as 50% increase in runtime from collision distance 0.05 to 1.5.
> >
> > Collision distance sensitivity for MeshGraphNets:
> > Metric |dc=0.05  | dc=0.1  | dc=0.5  | dc=1.0  | dc=1.5
> > --- |--- |--- |--- |--- |--- |
> > Translation error (m) | $ 0.777 \pm 0.050 $ | $ 0.652 \pm 0.063 $ | $ 0.174 \pm 0.012 $ | $ 0.126 \pm 0.006 $ | $ 0.127 \pm 0.013 $
> > Rotation error (deg) | $ 31.572 \pm 0.675 $ | $ 30.406 \pm 1.434 $ | $ 18.804 \pm 0.876 $ | $ 15.844 \pm 0.215 $ | $ 15.318 \pm 0.654 $
> > Runtime (s) | $ 0.233 \pm 0.021 $ | $ 0.226 \pm 0.015 $ | $ 0.288 \pm 0.048 $ | $ 0.264 \pm 0.014 $ | $ 0.321 \pm 0.008 $
> > Number of collision edges | $ 0.77 \pm 0.22 $ | $ 3.66 \pm 0.27 $ | $ 377 \pm 7 $ | $ 3311 \pm 4 $ | $ 10863 \pm 222 $
> >
> > > Finally, one of the weaknesses of the approach is the need to have access to a rich and detailed representation of the scene, in the form of a mesh. Have the authors considered solutions to train their models from weaker signals, or even directly from images?
> >
> > We agree that the model is limited to requiring states, as we point out in our limitations section. Extracting 3D meshes/particles from images / videos is an extremely interesting but also extremely challenging and open problem [1,2,3], which is out of scope for this paper. Notably, these techniques only currently work in mostly static environments (where objects are not moving). It would therefore be quite a challenging and interesting research problem to adapt them to the kinds of moving object videos we show here. We hope that our advances in learned simulators will be useful for later integration with a perceptual front-end.
> >
> > > In addition to the quality of the work offered, I note the use of certain tricks to keep the number of pages to the limit. In particular, the referencing of figures in the appendix in the main text is questionable, but I recognize that this is common practice. What concerns me more is the modification of the style of bibliography compared to the standards of ICLR. It seems to me that this style is mandatory (while it requires a lot of space). If this is indeed the case, I would appreciate if the authors regularize their paper, for the sake of fairness with other submissions.
> >
> > We apologize for the formatting. We based our choice on previous ICLR papers which have used this number-based bibliography formatting (e.g. MeshGraphNets (https://arxiv.org/pdf/2010.03409.pdf, and https://openreview.net/forum?id=FmBegXJToY). We therefore expected this formatting choice to be acceptable, given the existence of these prior works at ICLR using this formatting. Nevertheless we have changed the formatting to respect the (Name, year) style rather than numbers, with only very minimal changes to the text required to remain within the page limit. We hope this ameliorates the reviewer’s concern.
> >
> > [1] Gao, Jun, et al. "GET3D: A Generative Model of High Quality 3D Textured Shapes Learned from Images." arXiv preprint arXiv:2209.11163 (2022).
> >
> > [2] Byravan, Arunkumar, et al. "NeRF2Real: Sim2real Transfer of Vision-guided Bipedal Motion Skills using Neural Radiance Fields." arXiv preprint arXiv:2210.04932 (2022).
> >
> > [3] Li, Yunzhu, et al. "3d neural scene representations for visuomotor control." Conference on Robot Learning. PMLR, 2022.

---

> > > ### Comment · Reviewer_sivF · 2022-11-21
> > > **Thank you**
> > >
> > > I thank the authors for their very complete and convincing answers. The authors were able to answer each of my questions by providing many results and new experiments in a short time. The contribution now seems very convincing to me: although it is based on a relatively simple idea, ie. modeling interactions between faces, the results are very impressive, especially on a task as difficult as the one studied in the paper (I disagree with the reviewer edAB on this point, and look forward to seeing the evolution of this discussion).
> > >
> > > I also appreciate that the authors have aligned the layout with ICLR recommendations. I increased my score to the maximum rating, and think that this paper deserves to be presented in detail at ICLR.
> > >
> > > I indeed forgot to quote reference [1]:
> > > Milano, F., Loquercio, A., Rosinol, A., Scaramuzza, D., & Carlone, L. (2020). Primal-dual mesh convolutional neural networks. Advances in Neural Information Processing Systems.
> > > But I admit that the comparison with this reference does not necessarily constitute an important added value to the related works.
> > >
> > > PS: There is a margin overflow on page 16 of the appendix which could be corrected in the final revision of the paper.

---

### Official Review · Reviewer_zfn2 · 2022-10-23

**Confidence:** 4
**Correctness:** 4
**Technical Novelty And Significance:** 4
**Empirical Novelty And Significance:** 4
**Recommendation:** 10

**Clarity, Quality, Novelty And Reproducibility:**

Clarity: Excellent! The paper ie very well-written and a delight to read. The supplementary section is extensive and thorough and the results provided are numerous and compelling.

Novelty: Overall the work is highly original both in solving a previously unsolved problem and in proposing a novel architecture to solve it.

Quality: The quality of the presentation, the technical soundness of the work, the thoroughness of the experiments are all excellent.

Reproducibility: While the authors have not provided code or promised to do so, sufficient details of the experiments are provided in the supplementary section to enable reproduction of the work.

**Strength And Weaknesses:**

Strengths:
Novelty: The work solves the previously unsolved problem of how to enable effective and efficient message passing between colliding mesh faces in a GNN. To do so they introduce a task-specific module of encoding "Face-face edge features" along with mechanism for effective message passing between these new features and the existing nodes and edges of the graph. The proposed design is conceptually correct and well-motivated. The authors also (potentially) introduce a virtual object node at the center of each object to pass messages between distant nodes more efficiently. This too makes sense.

Clarity: The paper is well-written and the material is presented in a well-organized fashion.

Significance: The method shows impressive performance improvements over the SOTA methods in terms of accuracy and computational efficiency and generalization, outperforming many previous learning-based and analytical baselines. The authors' results on effectively learning real-world dynamics besides the ones from simulation are exciting and can open the door to many new applications.

Weaknesses:
1. It is a bit unclear from the text and experiments if the virtual object node is the author's novel contribution or it comes from prior works on particle based approaches. It would be good if the authors could state this more clearly in the text.

2. The authors have clearly acknowledged several limitations of their presented technique in terms of not being able to deal with stochastic outputs, requiring state information as opposed to predicting from images alone, and in terms of learning with even more complex objects with many orders of magnitude more points. I am satisfied with with this.

**Summary Of The Paper:**

This paper proposes a “Face Interaction Graph Network” (FIGNet)" architecture for accurately and efficiently modeling collisions between faces of meshes. Previous SOTA methods based on Graph Neural Networks (GNNs) model only collisions between nodes of the meshes of different objects and hence cannot deal effectively and efficiently with objects with large faces without requiring significant sub-division of the mesh's surfaces. There are two key innovations that the authors introduce into GNNs to solve this problem. The first is to introduce a mechanism for message passing between the face and the nodes that comprise it and the second is to introduce and object-level virtual node for efficient passing of information between nodes without requiring too many message passing iterations. The authors compare their approach to SOTA learning-based graph and particle methods for this task on a simulated dataset and against an analytical solver on a real-world object pushing task. They show SOTA performance in terms of accuracy, computational efficiency and generalizability of the proposed method versus all existing approaches.

**Summary Of The Review:**

Overall this paper introduces a significant and broadly impactful novel contribution to the computer vision research community, which can potentially open the doors to new research ideas and applications. Hence, I recommend accepting this work.

---

> ### Author Response · Authors · 2022-11-18
> **Response**
>
> Thank you for the thoughtful review! We are very glad you enjoyed the paper. We respond to your comments below.
>
> > It is a bit unclear from the text and experiments if the virtual object node is the author's novel contribution or it comes from prior works on particle based approaches. It would be good if the authors could state this more clearly in the text.
>
> We have updated the text to explain this. Using virtual nodes to facilitate rapid message-passing across a graph is not unique to this work. For example, see [1, 2] which use hierarchically represented virtual nodes based on clustering through *volumes* of particles. Because these prior methods used point clouds to represent objects instead of meshes, hierarchical clusters were used to reduce the computational cost of running the methods, as passing messages across all particles would be prohibitively expensive. However, to our knowledge, we are the first to propose a *single object node* that communicates with nodes representing *only* the surface of the object. As shown, this is particularly important not for computational efficiency, but rather for *accuracy* especially for objects with more complicated shapes.
>
> [1] Damian Mrowca, Chengxu Zhuang, Elias Wang, Nick Haber, Li F. Fei-Fei, Josh Tenenbaum, and Daniel L. Yamins. "Flexible neural representation for physics prediction." Advances in neural information processing systems 31 (2018).
>
> [2] Yunzhu Li, Jiajun Wu, Russ Tedrake, Joshua B. Tenenbaum, and Antonio Torralba. Learning particle dynamics for manipulating rigid bodies, deformable objects, and fluids. In International Conference on Learning Representations, 2019.

---

> > ### Comment · Reviewer_zfn2 · 2022-11-28
> > **Response to Authors**
> >
> > I thank the authors for clearly responding to my question and for updating the paper appropriately. I maintain that this paper makes a significant and insightful contribution to the research in this field and should be highlighted at the conference. I also tend to respectfully disagree with edAB's opinion that the paper considers a simple task. I will maintain my original rating for this paper.

---

### Official Review · Reviewer_dbQg · 2022-10-25

**Confidence:** 4
**Correctness:** 3
**Technical Novelty And Significance:** 3
**Empirical Novelty And Significance:** 2
**Recommendation:** 8

**Clarity, Quality, Novelty And Reproducibility:**

The presentation of the paper is clear and easy to follow. The idea of adding a message-passing mechanism between faces and the proposal of the virtual object node are novel to me.

**Strength And Weaknesses:**

**Strength:**

- The illustrations are helpful in understanding the idea, and the technical details are complete.

- This extension to MeshGraphNets is elegant.


**Weaknesses:**

- The following reference may be closely related to the topic:

*Lei Lan, et al. 2022. Affine Body Dynamics: Fast, Stable and Intersection-free Simulation*

The above simulator can achieve competitive or faster rigid-body dynamics than Bullet, Mujoco, and PhysX, and can guarantee non-penetration for arbitrarily complex shapes. I would like to see a wall-clock time cost comparison to Bullet, Mujoco, and PhysX as well.

- There are only two contact scenarios for discrete mesh: point-triangle contacts and edge-edge contacts, as shown in the following reference:

*Minchen Li, et al. 2020. Incremental Potential Contact: Intersection- and Inversion-free, Large-Deformation Dynamics.*

The experiments would be more thorough if the authors could specifically examine how these two contacts are resolved.

- Mathematically, contact is just a non-penetration constraint for simulation. I would like to see **quantitative** statistics on penetration counts or penetration depths when making comparisons.


**Summary Of The Paper:**

The paper introduces a novel face interaction mechanism for GNN-based rigid body simulators to solve the penetration issue existing when the contact point is far from nodes. The new framework significantly increases the accuracy of contact resolution and enables the simulation of complex shapes. The framework is even more expressive than analytical simulators. It can capture subtle dynamic behaviors shown in real-world data that cannot be interpret by some existing rigid-body simulators.

**Summary Of The Review:**

The paper proposes an extension to MeshGraphNets, which significantly increases the contact resolution accuracy. However, quantitative statistics about the efficiency and accuracy of the proposed framework as a rigid body simulator are needed.

---

> ### Author Response · Authors · 2022-11-18
> **Response to Reviewer**
>
> Thank you for the thoughtful review. We respond to your comments below.
>
> > Lei Lan, et al. 2022. Affine Body Dynamics: Fast, Stable and Intersection-free Simulation
>
> > Minchen Li, et al. 2020. Incremental Potential Contact: Intersection- and Inversion-free, Large-Deformation Dynamics.
>
> Thank you for pointing out these papers! The second appears to focus on elastic and soft bodies, but we have added the first as a citation in the introduction for papers which have pointed out the challenges of rigid body simulation. Please see the updated paper introduction. We would be happy to expand on connections further if the reviewer desires.
>
> > I would like to see a wall-clock time cost comparison to Bullet, Mujoco, and PhysX as well.
>
> We apologize if the exposition was unclear – we are not claiming to be faster than Bullet, Mujoco or PhysX. Rather, we are claiming to be faster than alternative *learned simulators* that represent contact dynamics only between nodes rather than considering contacts between faces. Relative to Bullet and Mujoco, we are instead claiming to be more accurate for real data, which we show in Figure 7. Our system is not optimized for speed. We agree it would be interesting to explore how to even further speed up these learned simulators so that they are competitive in wall clock time with analytic models. For example, follow up work could investigate various methods to simplify the contact meshes, as well as faster algorithms for distance calculation between faces. However, this would involve a significant amount of work that is outside the scope of this paper. We also provide measures of the wall clock time in the updated supplement in table I.1. For the reviewer’s convenience, we also include that table here. Note that this is approximately 10x slower than PyBullet, but again, we are not claiming to be faster than highly optimized analytic simulators, only more accurate for real world data. However, through this performance runtime comparison, we do see that FIGNet is significantly faster than alternative learned simulators.
>
> Model | Runtime (seconds)
> --- | ---
> FIGNet | $ 0.094 \pm 0.005  $
> MGN-LargeRadius+ | $ 0.258 \pm 0.010 $
> DPI-Reimplemented* | $ 0.126 \pm 0.006 $
> MGN+ | $ 0.160 \pm 0.007 $
>
> > There are only two contact scenarios for discrete mesh: point-triangle contacts and edge-edge contacts
>
> We agree that vertex-face and edge-edge tests are sufficient to detect all mesh-mesh contact scenarios, and it is a popular method used in real time physics engines. Indeed, our face-face collision test uses these procedures to determine whether two faces are colliding. We just parameterize the collision as face-face rather than considering vertex-face and edge-edge separately. This is simpler from a deep network parameters perspective (only requiring one new type of edge instead of separately parameterizing two new types of edges), but in principle vertex-face and edge-edge would also be possible. In response to your next point, we provide an analysis of face-face penetration distances and counts, which will capture all possible vertex-face and edge-edge collisions given the implementation of face-face distances.
>
> > Mathematically, contact is just a non-penetration constraint for simulation. I would like to see quantitative statistics on penetration counts or penetration depths when making comparisons.
>
> We have now provided these metrics for the FIGNet and MeshGraphNet models. To calculate penetrations and penetration distances, we use the algorithm from [1]. Note that this algorithm is imperfect for small faces, which some of our objects contain. We therefore ran penetration measures on both the ground truth dynamics from PyBullet, as well as for the learned models. We do not compare to DPI as it does not have faces (it is a particle-based method), so the algorithm from [1] will not apply. Given the worse translation and rotation errors though, we expect its penetration performance would also be worse than FIGNet. We report the penetration distance and penetration counts for the learned models as a ratio between the model and the ground truth for each trajectory, and report the mean across trajectories. This has also been added to the appendix in Table H.1.
>
> Model | Penetration distance ratio | Penetration count ratio
> --- | --- | ---
> FIGNet | $ 1.037 \pm 0.033 $ | $ 1.041 \pm 0.037 $
> MGN-LargeRadius+ | $ 1.071 \pm 0.020 $ | $ 1.073 \pm 0.018 $
> MGN+ | $ 4.613 \pm 0.143 $ | $ 5.246 \pm 0.187 $
>
> [1] J. A. Baerentzen and H. Aanaes, "Signed distance computation using the angle weighted pseudonormal," in IEEE Transactions on Visualization and Computer Graphics, vol. 11, no. 3, pp. 243-253, May-June 2005, doi: 10.1109/TVCG.2005.49.

---

> > ### Comment · Reviewer_dbQg · 2022-12-06
> > **Thanks for the response; Score updated**
> >
> > I thank the authors for addressing my concerns. I think this work is a useful extension to GNN-based neural simulators. I updated the score to 8.

---

### Official Review · Reviewer_edAB · 2022-10-25

**Confidence:** 4
**Clarity, Quality, Novelty And Reproducibility:** The paper is good in terms of the cla…
**Correctness:** 3
**Technical Novelty And Significance:** 3
**Empirical Novelty And Significance:** 3
**Recommendation:** 6

**Strength And Weaknesses:**

Strength

1.	The paper is well written so that it is easy to follow and understand the motivation and the proposed method.

2.	The proposed FIGNet is novel and experimentally demonstrated to be effective.

Weakness

1.	The experimental scenarios are a little simple, it would be better to conduct experiments on more realistic data.


**Summary Of The Paper:**

The paper proposes a Face Interaction Graph Network (FIGNet) to extend GNN-based rigid object collisions simulators for interaction computation between mesh faces, rather than nodes.  The proposed FIGNet is 4x more accurate in simulating complex shape interactions and 8x more computationally efficient on sparse and rigid meshes than learned node- and particle-based methods.

**Summary Of The Review:**

Based on the strength mentioned above, my current decision is accept.

---

> ### Author Response · Authors · 2022-11-18
> **Response to reviewer**
>
> Thank you for the review. We respond to your points below.
>
> > The experimental scenarios are a little simple, it would be better to conduct experiments on more realistic data.
>
> Could the reviewer elaborate on which aspects of the experimental scenario are simple, and what kind of data you would like us to train on instead? As far as we know, this is the most complex rigid dynamics data that has been simulated with a learned simulator, with objects that have 100s to 1000s of nodes in their meshes, and up to 12 objects interacting at a time. By contrast, one of the most recent works trained the simulator on data from only single objects interacting with the floor [1]. We further provide results on real world data from robotics in Figure 7, showing that we outperform analytic simulators even in the low-data regime.
>
> [1]  Kelsey R Allen, Tatiana Lopez Guevara, Yulia Rubanova, Kim Stachenfeld, Alvaro SanchezGonzalez, Peter Battaglia, and Tobias Pfaff. Graph network simulators can learn discontinuous, rigid contact dynamics. In 6th Annual Conference on Robot Learning, 2022.

---

### Author Response · Authors · 2022-11-18
**Summary of response**

We would like to thank all reviewers for their thoughtful reviews. We provide here a summary of changes to the paper and additional experiments.

**Additional experiments**
* As asked for by Reviewer dbQg, we calculated penetration distances for all models on the Movi-A dataset, showing that FIGNet maintains low penetration distances and counts relative to the ground truth model it was trained on (Section H, Table H.1).
* As asked for by Reviewer sivF, we ran a sweep over different collision distances for the MeshGraphNets model to show that increasing the collision distance does exponentially increase the number of collision edges empirically, as we suggested it should theoretically (Section J).
* As asked for by reviewers dbQG and sivF, we measured runtime in seconds for all models, and have included the results as Table I.1.
* As asked for by reviewer sivF, we tested the sensitivity to collision distance for the FIGNet model. These results are in section J.1.

**Changes to the paper**
* We updated the references to include suggestions from Reviewer dbQg
* We changed all bibliographic formatting to use (Name, year) format as requested by Reviewer sivF.
* We made other small clarifying changes asked for by some reviewers.
* We added results tables for the Kubric datasets to the appendix in Section K.

---

### Decision · Program_Chairs · 2023-01-20

**Decision:**

Accept: notable-top-25%

**Justification For Why Not Higher Score:**

This paper would be much stronger to evaluate if this framework can be used for physical reasoning tasks [1,2,3]. Ac would urge the authors to include some discussions in the camera-ready version.

[1] Physion: Evaluating Physical Prediction from Vision in Humans and Machines. NeurIPS 2021
[2] CLEVRER: CoLlision Events for Video REpresentation and Reasoning. ICLR 19
[3] ComPhy: Compositional Physical Reasoning of Objects and Events from Videos. ICLR 22

**Justification For Why Not Lower Score:**

Novel framework and strong results!

**Metareview: Summary, Strengths And Weaknesses:**

Four experts reviewed this paper with all accepted recommendations. The area chairs agree that this work makes a very important contribution by introducing a novel Face Interaction Graph Network (FIGNet)  to significantly improve the SOTA of learning rigid body dynamics. The reviewers did raise some valuable concerns that should be addressed in the final camera-ready version of the paper.




**Note From Pc:**

if the above contains the word "oral" or "spotlight" please see: "oral" presentation means -> notable-top-5% and "spotlight" means -> notable-top-25%. As stated in our emails, we are disassociating presentation type from AC recommendations